# Feedback mechanisms controlling Antarctic glacial cycle dynamics simulated with a coupled ice sheet–solid Earth model

Torsten Albrecht [1], Meike Bagge [2,3], and Volker Klemann [2]

[1]Earth Resilience Science Unit, Potsdam Institute for Climate Impact Research (PIK), Member of the Leibniz Association, Potsdam, Germany
[2]Helmholtz Centre Potsdam, German Research Centre for Geosciences - GFZ, Potsdam, Germany
[3]now at: Federal Institute for Geosciences and Natural Resources, Hanover, Germany

**Correspondence:** T. Albrecht (albrecht@pik-potsdam.de)

**Abstract.** The dynamics of the ice sheets on glacial time scales are highly controlled by interactions with the solid Earth, i.e., the glacial isostatic adjustment (GIA). Particularly at marine ice sheets, competing feedback mechanisms govern the migration of the ice sheet's grounding line (GL) and hence the ice sheet stability. For this study, we developed a coupling scheme and performed a suite of coupled ice sheet–solid Earth simulations over the last two glacial cycles. To represent ice sheet dynamics we apply the Parallel Ice Sheet Model PISM and to represent the solid Earth response we apply the three-dimensional viscoelastic Earth model VILMA, which, in addition to load deformation and rotation changes, considers the gravitationally consistent redistribution of water (the sea-level equation). We decided on an offline coupling between the two model components. By convergence of trajectories of the Antarctic Ice Sheet deglaciation we determine optimal coupling time step and spatial resolution of the GIA model and compare patterns of inferred relative sea-level change since the Last Glacial Maximum with the results from previous studies. With our coupling setup we evaluate the relevance of feedback mechanisms for the glaciation and deglaciation phases in Antarctica considering different 3D Earth structures resulting in a range of load-response time scales. For rather long time scales, in a glacial climate associated with far-field sea level low stand, we find GL advance up to the edge of the continental shelf mainly in West Antarctica, dominated by a self-amplifying GIA feedback, which we call the 'forebulge feedback'. For the much shorter time scale of deglaciation, dominated by the Marine Ice Sheet Instability, our simulations suggest that the stabilizing sea-level feedback can significantly slow-down GL retreat in the Ross sector, which is dominated by a very weak Earth structure (i.e. low mantle viscosity and thin lithosphere). This delaying effect prevents a Holocene GL retreat beyond its present-day position, which is discussed in the scientific community supported by observational evidence at the Siple Coast and by previous model simulations. The applied coupled framework, PISM-VILMA, allows for defining restart states to run multiple sensitivity simulations from. It can be easily implemented in Earth System Models (ESMs) and provides the tools to gain a better understanding of ice sheet stability on glacial time scales as well as in a warmer future climate.

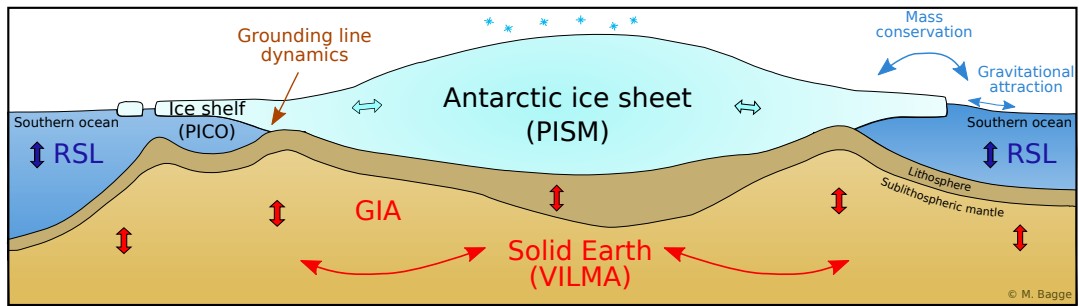

**Figure 1. Schematics of coupled PISM Ice Sheet – PICO Ice Shelf – VILMA Solid Earth model components**. Bedrock topography, relative sea level (RSL) and ice thickness are exchanged between VILMA and PISM in predefined coupling time steps.

## 1 Introduction

The observed accelerating retreat of large sectors of the Antarctic Ice Sheet (AIS) raises concerns about future sea-level rise (Seroussi et al., 2020; IPCC, 2021). Self-amplifying, positive feedbacks may trigger the tipping and hence the destabilization and collapse of the marine-based parts of the ice sheet (Pattyn and Morlighem, 2020; Armstrong McKay et al., 2022; Lenton et al., 2023). The Marine Ice Sheet Instability (MISI) is mainly controlled by the bedrock geometry (Fretwell et al., 2013; Morlighem et al., 2020), with retrograde slopes supporting a run-away dynamic feedback mechanism (Weertman, 1974; Mercer, 1978; Thomas and Bentley, 1978; Schoof, 2007), while buttressing forces can have a stabilizing effect for certain ice-shelf geometries (Gudmundsson, 2013; Pegler, 2018; Haseloff and Sergienko, 2018; Reese et al., 2018b). The dynamic retreat of ice sheet grounding lines in response to the thinning of ice shelves and the subsequent speed-up of ice streams mainly in West Antarctica have been the dominant drivers of most of the recent mass loss in Antarctica (Rignot et al., 2019; Otosaka et al., 2023), and some studies suggest that MISI may already be underway (Joughin et al., 2014; Rignot et al., 2014; Reed et al., 2023). Depending on the water depth at marine-terminating glacier fronts, the ice sheet retreat may also be amplified by the Marine Ice Cliff Instability (MICI) (Pollard et al., 2015; DeConto and Pollard, 2016; Crawford et al., 2021; Li, 2022).

The solid Earth responds to ice sheet thinning or retreat, or more general to loading processes, by viscoelastic deformation, generally known as glacial isostatic adjustment (GIA), see Fig. 1. The strength as well as typical length and time scales of the GIA depend on the Earth's mantle viscosity and lithosphere thickness (e.g. Peltier, 1974). The consistent redistribution of land ice, ocean water and mantle material induces changes in the Earth's gravity field (with the 'geoid' as an equipotential surface), and therefore alters the sea level globally, which follows the geoid (Farrell and Clark, 1976). As a further feedback mechanism, polar motion due to the surface mass redistribution changes the geoid and deforms the solid Earth (Mitrovica et al., 2005). This gravitational-rotational-deformational (GRD) correction is usually considered as being part of the GIA.

Geodetic and geophysical investigations of crustal uplift in the Amundsen Sea sector in West Antarctica (Barletta et al., 2018; Lloyd et al., 2020; Blank et al., 2021) postulate a laterally strongly varying viscoelastic Earth structure with mantle viscosities orders of magnitude lower than the global average. Hence, a low mantle viscosity implies much faster and, in the likely case of an also thinner lithosphere, a more localised (short-wavelength) response of the solid Earth to ice sheet retreat (van der Wal

et al., 2015; Larour et al., 2019; Wan et al., 2022). Furthermore, geophysical investigations demand a clear dichotomy in the Earth structure between East and West Antarctica (Behrendt, 1999; Morelli and Danesi, 2004; Accardo et al., 2014; Lloyd et al., 2020; Powell et al., 2020). But also within West Antarctica a large degree of heterogeneity exists on length scales of 100 km (An et al., 2015; Ramirez et al., 2016; Heeszel et al., 2016), with hot spots of enhanced geothermal heat flux (Lough et al., 2013; Dziadek et al., 2021; Bredow et al., 2023).

The relative sea level (RSL), which is the water depth when positive or more precisely the vertical distance between the geoid and the bottom of the ocean, directly determines the grounding line (GL) position of marine ice sheets via the flotation criterion. A global mean (barystatic) sea level can be calculated from the global RSL change, or by aggregating changes in ice sheet thickness and near-field RSL changes, based on the 'volume above flotation' (Gregory et al., 2019; Goelzer et al., 2020; Adhikari et al., 2020). When ice sheets decay and GLs retreat, the viscoelastic solid Earth deformation reduces the water depth and hence induces a stabilizing negative feedback, which is additionally supported by near-field sea level lowering in response to reduced gravitational attraction (Gomez et al., 2010, 2012, 2015; Coulon et al., 2021; Han et al., 2021).

As a side effect, the vertical bed displacement induces a bending of the elastic lithosphere and lateral viscous flow of mantle material, which is associated with the migration of a bulge with opposite RSL sign in the vicinity (Adhikari et al., 2014; Wan et al., 2022). Accordingly, local surface motion and slopes may changes sign during lateral motion of the topographic bulge, which makes it difficult to isolate accelerating or inhibiting effects on the ice flow. During deglacial or interglacial periods, the 'forebulge' subsidence is often associated with a mechanism termed 'ocean syphoning' mainly in view of the northern-hemisphere deglaciation (Mitrovica and Peltier, 1991; Mitrovica and Milne, 2002), which draws water from the far field and thereby contributes a far-field sea-level fall (Powell et al., 2021; Yousefi et al., 2022). The latitudinal extent of the forbulge region strongly depends on the lithosphere thickness. It can be either confined close to the coast and inner continental shelf for a thin lithosphere or reach offshore for thick lithospheres (Stocchi et al., 2013).

In grounded ice sheet regions, where bedrock uplift directly affects the ice sheet's height, the surface climate becomes generally colder and dryer (lapse-rate effect), which in turn affects the overall ice dynamics (Van den Berg et al., 2008; Konrad et al., 2014; Zeitz et al., 2022). Potentially, solid-Earth feedbacks can slow down or even halt deglacial retreat, depending on the time scales of the viscoelastic deformation controlled by the structure of the solid Earth (Gomez et al., 2010; Konrad et al., 2015; Larour et al., 2019). In cooling phases with GL advance, i.e. during glaciation, coupled model studies suggest a smaller glacial extent (and ice volume) than in stand-alone ice sheet models (forced with global mean sea level), as self-gravitational attraction and local RSL changes dominate, associated with this negative (self-stabilizing) feedbacks of the RSL (Gomez et al., 2013; de Boer et al., 2014, 2017). This complex interplay of solid Earth, sea level and ice dynamics has therefore played a major role during glacial cycles (Whitehouse et al., 2019) and is relevant for interpreting constraints on GIA in the periphery of the ice sheet, such as sea-level indicators and GPS uplift rates (Konrad et al., 2014).

A global sea-level low stand was reached during the Last Glacial Maximum (LGM) between 26,500 and 19,000 years ago (26.5–19 kyr BP, Clark et al., 2009). The rapid sea-level rise since LGM due to the collapse of the northern hemisphere ice sheets likely contributed to the destabilisation of the AIS, as iceberg-rafted debris records in deep-sea sediments suggest

(Weber et al., 2014; Jones et al., 2022). Accordingly this is reflected in an inter-hemispheric synchronicity of the large ice sheets' changes (Gomez et al., 2020).

There is growing evidence that during the Holocene period (the past 11,700 years) the solid-Earth uplift in combination with ice rise formation may have caused the re-advance of the Ross Sea GL to present-day position after an initial phase of rapid deglaciation (Lingle and Clark, 1985; Kingslake et al., 2018; Johnson et al., 2022; Lowry et al., 2024), while in the
Amundsen Sea GLs remained stable for the last 5,500 years until recently (Braddock et al., 2022). The present-day state of the AIS, including its contemporary rates of change, characterizes its stability and the potential for sea-level rise in an increasingly warming climate (Joughin and Alley, 2011).

Ice sheet models have been coupled to GIA models with different levels of complexity (de Boer et al., 2017; Whitehouse, 2018; Swierczek-Jereczek et al., 2024, see overview Table 1). The computationally efficient two-layer approach with an 'Elastic
Lithosphere and a Relaxing Asthenosphere' (ELRA) accounts for the vertical bedrock adjustment for one given relaxation time (Brotchie and Silvester, 1969; Le Meur and Huybrechts, 1996) and is widely used in ice sheet modeling or coupled Earth system modeling (Pollard and DeConto, 2012; DeConto and Pollard, 2016; Pattyn, 2017; Quiquet et al., 2018). The local response time and flexural regidity can be also varied laterally (Coulon et al., 2021).

ELRA can be improved when the time-lagged mantle flow in the asthenosphere below an elastic thin plate is approximated by
solving the corresponding biharmonic differential equation using Fast Fourier Transformation (Lingle and Clark, 1985; Bueler et al., 2007; Albrecht et al., 2020a; Book et al., 2022). In this 'Lingle-Clark' (LC) bed deformation model, a wavelength-depending response time spectrum replaces the single exponential decay parameter of the ELRA approximation. Hence, only the uniform mantle viscosity and flexural rigidity (or elastic lithosphere thickness) are used as input parameters valid for the considered half-space.

In addition to the bedrock changes, global mean sea level anomalies are used as external forcing for ice sheet models to account for water depth changes at the GL (Goelzer et al., 2020). The LC model is limited to a certain region (e.g. Antarctica) and is unable to prescribe gravitational, globally self-consistent water-load changes or to account for feedbacks associated with polar motion due to GIA. The computationally inexpensive approach has been recently extended by considering lateral variations in Earth structure and a regional sea-level change (Swierczek-Jereczek et al., 2024).

In order to account for both global and near-field spatial variability of relative sea-level change consistent with dynamic changes in ice sheet extent, coupled ice sheet–solid Earth models need to simultaneously solve the 'sea-level equation' (Farrell and Clark, 1976), as realized in 'Self-Gravitating Viscoelastic solid-Earth Models' (SGVEM: e.g., de Boer et al., 2014; Gomez et al., 2015, 2020; Konrad et al., 2015; Han et al., 2022). GIA models generally account for both solid-Earth deformation and gravitational effects in combination with rotational effects in response to changes in the redistribution of ice and ocean masses
(Milne and Mitrovica, 1996).

Most GIA models so far compute changes in Earth deformation and geoid for a radially varying (depth dependent) 1D Earth structure using spherical harmonics (Whitehouse et al., 2012; Konrad et al., 2014, 2015; Pollard et al., 2017). In regions of weak Earth structure, as for instance in West Antarctica, the observed uplift rates are not compatible with the response of a

viscosity structure usually applied for northern-hemisphere GIA in the range of $10^{20}$–$10^{21}\,\mathrm{Pa\,s}$ and a lithosphere thickness of
$\approx 100\,\mathrm{km}$ (Whitehouse, 2018; Ivins et al., 2023).

Since the 2000s, 3D viscoelastic Earth models were developed (e.g. Martinec, 2000; Latychev et al., 2005; Wu et al., 2005; Zhong et al., 2022; Huang et al., 2023) and considered in a new generation of GIA models (e.g. Klemann et al., 2008; A et al., 2012; van der Wal et al., 2015; Nield et al., 2018; Bagge et al., 2021; Blank et al., 2021; Yousefi et al., 2022). Interactively coupled to ice sheet models, such 3D GIA models find significant differences in West Antarctic Ice Sheet (WAIS) evolution
on glacial time scales compared to 1D GIA or standalone ice sheet models (Gomez et al., 2018; Han et al., 2022; Van Calcar et al., 2023; Swierczek-Jereczek et al., 2024). The underlying coupling methods all imply a similar iterative process to account for the unknown initial bed topography, but differ with respect to the coupling time steps ranging from hundreds to thousands of years.

Here, we present a set of new simulations of AIS evolution over the last 246,000 years (i.e. two full glacial cycles) with the
Parallel Ice Sheet Model (PISM) coupled to the VIscoelastic Lithosphere and MAntle model (VILMA) solving for GIA.

## 2   Methods

### 2.1   PISM

The Parallel Ice Sheet Model (PISM User Manual, 2023, here based on v1.2.2, see also Section 'code and data availability'), is an open-source three-dimensional ice sheet model (Winkelmann et al., 2011; The PISM authors, 2023), written in C++
programming language. PISM has been previously applied in glacial-cycle simulations (e.g. Golledge et al., 2014; Sutter et al., 2019; Albrecht et al., 2020a, b), and can be easily coupled to other climate or Earth system model components (e.g. Kreuzer et al., 2021; Ziemen et al., 2019; Yang et al., 2022).

We here use a hybrid combination of two stress-balance approximations for the deformation of the ice, the Shallow Ice – Shallow Shelf Approximation (SIA-SSA, Hindmarsh, 2004), that guarantees a smooth transition from vertical-shear dominated
flow in the interior via sliding-dominated ice stream flow to fast plug flow in the floating ice shelves (Bueler and Brown, 2009), while neglecting higher-order modes of the flow. Driving stress at the GL is discretized using one-sided differences (Feldmann et al., 2014). The GL position simply results from flotation condition, without additional flux conditions imposed (Reese et al., 2018c). Basal friction and basal melt are interpolated according to a linear sub-grid interpolation scheme (Gladstone et al., 2010; Seroussi and Morlighem, 2018; Bradley and Hewitt, 2024). Thus, GL migration is reasonably well represented in PISM
(compared to Stokes flow), even for coarse resolution (Pattyn et al., 2013; Feldmann et al., 2014).

Ice deforms according to the Glen-Paterson-Budd-Lliboutry-Duval flow law with exponent $n = 3$ (Lliboutry and Duval, 1985). PISM simulates the three-dimensional polythermal enthalpy conservation for a given surface temperature and basal heat flux and thus captures melting and refreezing processes in temperate ice (Aschwanden and Blatter, 2009; Aschwanden et al., 2012). It is solved on a 3-dimensional grid with 81 vertical layers, with $20\,\mathrm{m}$ resolution at the base. The energy con-
servation scheme also accounts for the production of sub-glacial (and transportable) water (Bueler and van Pelt, 2015), which affects basal friction via the concept of a saturated and pressurized sub-glacial till. The till friction angle, which accounts for

characteristics of the underlying substrate, has been optimized for present-day ice flow (Albrecht et al., 2020a). Depending on the resultant yield stress this allows for grow-and-surge instability (Feldmann and Levermann, 2017; Bakker et al., 2017; Schannwell et al., 2023). Here we use the non-conserving hydrology model, where the till water content in each grid cell is balanced by basal melting and a constant drainage rate.

Ice shelf margins can evolve up to the edge of the continental shelf (here defined at 1800 m depth) constrained by a terminal thickness criterion (>75 m). Iceberg formation from ice shelves is parameterized based on spreading-rates (Levermann et al., 2012). Ice shelf melting is calculated using the Potsdam ice shelf Cavity mOdel (PICO) that considers basin-mean ocean properties on the continental shelf in front of the ice shelves and mimics the vertical overturning circulation in the ice shelf cavity (Reese et al., 2018a).

PISM comes with a generalized version of the Lingle-Clark bedrock deformation model (Bueler et al., 2007), assuming an elastic lithosphere, a resistant asthenosphere and a viscous half-space below (Whitehouse, 2018). The LC model is computationally efficient, the default coupling time step is 10 yr, but it does not account for regional sea-level variations due to gravitational attraction of the ice as of the deforming solid Earth.

A continental-scale representation of modern Antarctic bed and ice sheet topography is obtained from the Bedmap2 dataset (Fretwell et al., 2013). We simulate the entire Antarctic continent with 16 km grid resolution (381×381 regular grid size) compatible with the definition of the initMIP and ISMIP6 model intercomparison project (Nowicki et al., 2016, 2020). With an adaptive explicit time stepping of around $0.34 \pm 0.03$ yr, one glacial cycle (123 kyr) can be simulated within 1–2 days wall clock time with 64 CPU cores on a standard memory computing node at DKRZ (see https://docs.dkrz.de/doc/levante/configuration.html, last access: 05 June 2024, 'AMD 7763' with 128 CPU cores in total, 256 GB main memory and 2.45 GHz base clock).

## 2.2 VILMA

The VIscoelastic LIthosphere and MAntle model (VILMA) solves the field equations for an incompressible self-gravitating viscoelastic sphere in the Lagrange domain, laterally in spherical harmonics, radially with finite elements, and in time using an explicit time-differencing scheme assuming a Maxwell rheology as material law. Lateral variations in viscosity are considered as shear energy perturbations, which are calculated on a Gauss-Legendre grid in the spatial domain and which are transformed back into the spectral domain at each time step (for details see Martinec, 2000). The code has been benchmarked for a spherical symmetric Earth structure (Spada et al., 2011), i.e. for a 1D Earth structure, and was applied in studies accounting for lateral variations, i.e. for a 3D Earth structure (Klemann et al., 2008; Bagge et al., 2021). Furthermore, the effect of rotational feedback is accounted for (Martinec and Hagedoorn, 2014), following the discussion in Mitrovica et al. (2005).

The sea-level equation, describing the gravitationally consistent and mass-conserving redistribution of ice masses and ocean water (Farrell and Clark, 1976; Mitrovica and Milne, 2003; Kendall et al., 2006), is solved in the spatial domain (Hagedoorn et al., 2007) and was benchmarked in Martinec et al. (2018). Coastlines can freely migrate according to the local RSL. The loading effect of floating versus grounded ice is accounted for. The GL position is determined by the flotation conditions and hence by densities of ice and ocean water (consistent with PISM). The GL migration is associated with a redistribution of

water mass between ice sheets and ocean, which can result in large non-uniform near-field RSL patterns (Mitrovica et al., 2001; Spada et al., 2013).

For the radial discretization, we chose $\Delta z = 5\,\mathrm{km}$ down to $420\,\mathrm{km}$ depth, followed below by $\Delta z = 10\,\mathrm{km}$ down to $670\,\mathrm{km}$ and $\Delta z = 40$ to $60\,\mathrm{km}$ down to the core–mantle boundary, where the parameterisation of the elastic shear modulus and density structure follows the Preliminary Reference Earth Model (PREM, Dziewonski and Anderson, 1981). The interaction with the fluid core is considered as a boundary condition, confining the solution domain to the Earth's crust and mantle. The spectral resolution is set to Legendre degree and order 170, meaning that layers with lateral variations are discretized on a $256 \times 512$ (n128) grid corresponding to a wavelength of $\sim 120\,\mathrm{km}$. The grid size is based on an alias-free condition, which defines the numbers of latitudinal Gauss-Legendre points to be larger than 3/2 of the maximum Legendre degree considered (e.g. Martinec, 1989). We consider this as sufficient due to the general low-pass response of the Earth to surface loading and the spectral resolution of the considered tomographic model of only degree 63 (Steinberger, 2016). Higher resolutions were tested in this project, but did not show significant changes (see Sect. 3.1). The sea-level equation, defining the gravitationally-consistent mass redistribution between oceans and ice sheets in response to the deforming solid Earth, is solved in the spatial domain, also on a Gauss-Legendre grid, but with a significantly higher resolution of $1024 \times 2048$ (n512) corresponding to a wavelength of $\approx 30\,\mathrm{km}$.

In order to allow for restarts, the original Fortran77 code was modularized and in main parts extended using Fortran90 commands during the PalMod project (see acknowledgements), and OpenMP parallelization (OpenMP, 2012-2023) was implemented explicitly. In our coupled simulations, VILMA uses explicit time steps of $2.5\,\mathrm{yr}$, in order to be able to solve also for very low mantle viscosities at the order of $10^{19}\,\mathrm{Pa\,s}$. Considering this set up, one glacial cycle ($123\,\mathrm{kyr}$) can be simulated within $19\,\mathrm{h}$ wall clock time with $64\,\mathrm{CPU}$ cores on a standard memory computing node (same as indicated for PISM).

## 2.3 PISM-VILMA Coupling

PISM and VILMA are offline-coupled using a coupling time step that can be rather short and is only limited by the explicit time step chosen in VILMA and PISM. As default, we apply a coupling time step of $\Delta t = 100\,\mathrm{yr}$, at which PISM and VILMA exchange data. The strategy, where ice load and relative sea level are exchanged (see below), was already applied in studies with an older version of VILMA coupled with dynamic ice sheet models RIMBAY and PSUICE-3D (Konrad et al., 2014, 2015, respectively), and also in the Earth system model CLIMBER-X (Willeit et al., 2022), where the current VILMA version was coupled directly to SICOPOLIS as a fortran module. Here, for the first time the 3D feature of VILMA is applied, whereas in the former studies only a 1D Earth was considered.

In a first step, PISM runs for the coupling interval (from $t_0$ to $t_1$), based on the initial bed elevation, which remains constant over the coupling time step $\Delta t$ (see Fig. 2). Subsequently, VILMA integrates over the same interval (from $t_0$ to $t_1$) according to a global ice sheet load history, which combines ICE-6G_C (Peltier et al., 2015) north of 60°S and the PISM-simulated AIS changes south of 60°S, remapped to the VILMA grid. VILMA interpolates the ice sheet history between the snapshots at beginning and end of the coupling interval and provides the global change in RSL at predefined snapshots. The RSL at $t_1$ is (bilinearly) remapped back to the PISM grid.

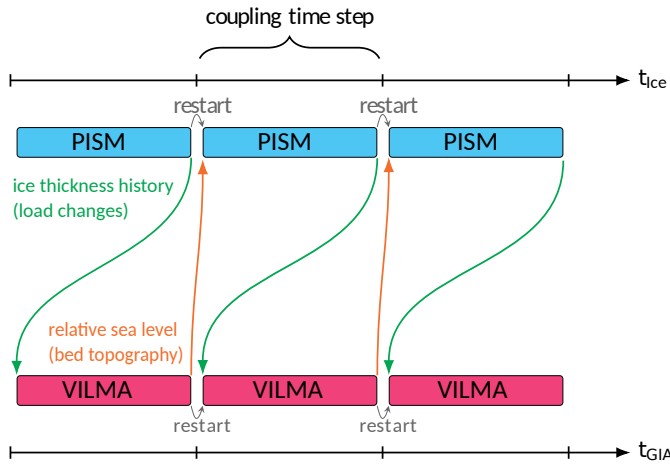

**Figure 2. PISM-VILMA coupling scheme**, adopted from Kreuzer et al. (2021).

Then, this process restarts for the next coupling interval ($t_1$ to $t_2$). PISM interprets the RSL change as negative vertical displacement of the bed elevation with respect to a reference surface elevation (geoid), or as change in water depth, if the bed elevation has negative values. That means, the surface elevation of the ocean in the PISM simulations remains at $z = 0$. With the adjusted bed topography (water depth), the ice sheet's GL can move directly due to the flotation criterion or indirectly due to induced ice dynamical changes. These changes in the ice load are then passed to VILMA again, which simulates the same coupling time step (from $t_1$ to $t_2$), before PISM simulates for the next coupling interval.

In our coupling scheme, the PISM response always lags behind the VILMA step by one coupling step, i.e. RSL change from $t_{i-1}$ to $t_i$ calculated in VILMA is only reflected in ice dynamics in PISM from $t_i$ to $t_{i+1}$. The implied numerical error remains small if short coupling time steps are used (see sensitivity to coupling time step in Sect. 3.1). We do not make use of (internal) iterations within a (longer) coupling time step as suggested, e.g., by Van Calcar et al. (2023), even though this could be easily implemented in PISM-VILMA. Furthermore, we avoid extrapolation of sea level and bed elevation changes to the next coupling interval as in Konrad et al. (2015), which principally can cause numerical instabilities. We also do not pass the sub-grid information of GL interpolation, used in PISM for the basal friction, to VILMA. In case of GL migration, this could avoid ad-hoc load changes from single grid cells. Our python coupler pismvilma (Albrecht, 2024b) uses CDO and NCO tools (CDO; NCO, 2003-2024) for remapping two-dimensional RSL changes from the global Gauss-Legendre grid used in VILMA to the Antarctic Polar Stereographic Maptiler (2024) equidistant Cartesian grid used by PISM and, vice versa, Antarctic ice thickness from the PISM grid back to the global VILMA grid.

On a High Performance Computing (HPC) system with a 'slurm' batch queue system, we make use of the heterogeneous job support (SLURM, 2003-2024), which can simultaneously optimize for the openMP parallelization (OpenMP, 2012-2023) on a single compute node (in case of VILMA and the python coupler) and the MPI parallelization that can be utilized across multiple compute nodes (in case of PISM). The remapping takes about 23% of computational time for 10-yr coupling time step, 9% for 100 yr and only 1.5% for 1000- yr, when using 3D Earth structures (see PISM2VILMA and VILMA2PISM in

| step | $100 \times 10$ yr | $10 \times 100$ yr | $1 \times 1000$ yr |
|---|---|---|---|
| PISM | 2876 (49.7%) | 1078 (52.2%) | 999 (60.3%) |
| PISM2VILMA | 871 (15.1%) | 141 (6.8%) | 19 (1.1%) |
| VILMA | 1581 (27.3%) | 802 (38.9%) | 632 (38.2%) |
| VILMA2PISM | 459 (7.9%) | 43 (2.1%) | 7 (0.4%) |
| TOTAL | 5788 (1.60 h) | 2063 (0.57 h) | 1657 (0.47 h) |

**Table 1. Mean wall clock time for PISM-VILMA steps** in seconds (and %) for 1 kyr simulation for the reference 3D Earth structure with 64 CPU cores and different coupling time steps, run on Levante (DKRZ). All three coupled simulations used the reference spatial resolution of n512 for the sea level equation in VILMA, n128 for the viscoelastic deformation in VILMA and 16 km for PISM in the Antarctic domain.

Table 1). Total computation time for simulating 1 kyr on a standard memory computing node (128 cores in total, see details in Sect. 2.1) is about 25 min for the 1000 yr coupling time step (25 CPUh), 30 min for 100 yr (35 CPUh) and 100 min for 10 yr (100 CPUh), when using 64 CPU cores for VILMA, PISM and the remapping. Considering only 1D (radial) Earth structures in VILMA brings a speed-up by a factor of 5 for the VILMA step and hence of almost 50% speed-up in the coupled model performance, for coupling time steps of 100 yr.

For comparison, Van Calcar et al. (2023) used a 3D GIA finite-element model with 16 CPU cores to simulate the last glacial cycle, with coupling time steps between 5000 and 500 yr, 40 km resolution in the ice sheet model and between 30 and 200 km in the GIA model, showed a performance of 16 CPUh per 1 kyr. However, when internal iterations were considered, the coupled model system required about 120 CPUh per 1 kyr.

In order to account for (long) memory effects of the AIS to the climate history, we run the coupled simulation for two full glacial cycles (246 kyr, see Fig. 3 a), with the northern hemisphere described by the ICE-6G_C reconstruction (available for the last 122 kyr, repeated for the penultimate glacial cycle with a time shift by 124 kyr and concateneted at the Last Interglacial), similar to Han et al. (2022). For consistency, the climatic and ocean forcings are applied as described in previous PISM standalone simulations (Albrecht et al., 2020a), which have been coupled to the LC bed deformation model (Bueler et al., 2007). In order to support an earlier deglaciation around 15 kyr BP (see Fig. 3 b) providing a better match with paleo proxies (Briggs and Tarasov, 2013; Albrecht et al., 2020b), PISM parameters were slightly adjusted in this study (e.g. till water drainage rate or precipitation scaling). Simulations are run until the year 1950 (corresponding to 0 kyr BP), such that the recent anthropogenic warming is ignored in the ice-core reconstruction based climate forcing (here a combination of EPICA Dome C and WAIS Divide ice core, see Albrecht et al., 2020a).

For the initial global bed topography we apply the present-day ETOPO1 bathymetry (Amante and Eakins, 2009), updated with Bedmap2 (Fretwell et al., 2013) in the Antarctic sector. As the initial bed topography of the simulation (at 246 kyr BP) is not known a priori, we iterate several times over the period of two glacial cycles (Kendall et al., 2005; Van Calcar et al., 2023), and successively correct for the initial bed topography according to the mismatch of the present-day bed topography to observations from Bedmap2 of the previous iteration (compare Fig. 3 and Fig. 4). Ideally, the present-day RSL subtracted

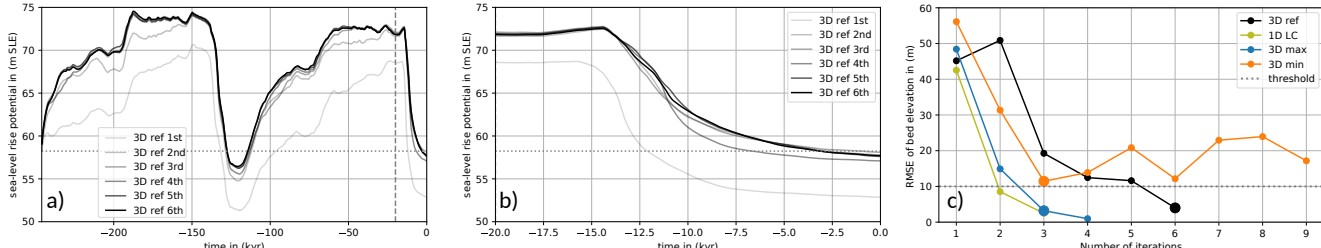

**Figure 3. Sea-level rise potential in coupled PISM-VILMA simulation** over a) 246 kyr and b) over the last 20 kyr showing convergence for six subsequent iterations with reference 3D Earth structure ('3D ref') at n128 resolution and n512 resolution for the sea-level equation. In each iteration the initial bed topography is adjusted, such that the misfit of present-day bed topography according to VILMA is minimized. c) Root-Mean-Squared Error of present-day bed topography anomalies in subsequent iterations for four different Earth structures, see Sect. 2.4 and Sect. 3.3. Big markers indicate last iteration selected for further comparison.

from the initial bed topography should converge to the present-day topography. In the first iteration, the present-day RSL is underestimated by up to 100 m in East Antarctica (bedrock too high) and overestimated by up to 100 m in West Antarctica (Fig. 4a). As measure of the mismatch we use the Root Mean Squared Error (RMSE) over the Antarctic computational domain. Generally, the RMSE reduces with each iteration by a factor of 0.2–0.4 (see Fig. 3 c). We stop iterating when the RMSE falls

below the acceptance level of 10 m (dotted line). This iterative procedure aims at minimizing deviations from the present-day bed topography, but we also find convergence of the present-day Antarctic ice volume against observations (Fig. 3 b); nevertheless in some regions the misfit in ice thickness remains comparably high (Fig. 4g–l). In order to convert Antarctic ice volume change in more practical units for better comparison, we here use the 'volume above flotation' approach with corrections for the density and in regions grounded below sea-level (Adhikari et al., 2020, Eq. 10). We apply this approach to

the projected regional ice model domain using cell-area weights and a constant ocean area of 362.5 million $\mathrm{km}^2$. Compared to an ice free state with present-day bed topography, this conversions provides a global mean estimate of the 'sea-level rise potential' (SLRP) in units of m SLE. As vertical land motion is already included in our RSL variable, we do not explicitly account for the 'water-expulsion effect' in the conversion (Goelzer et al., 2020; Pan et al., 2021).

## 2.4 Solid-Earth structures

Between the West and East Antarctic plates, differences in the upper-mantle viscosity, here shown at 280 km depth, can exceed two orders of magnitude (Fig. 5b). Based on seismic tomography models and geodynamical constraints, a dataset for the 3D Earth structure (including the lithosphere thickness) has been optimized such that the VILMA response minimizes the misfit to a global compilation of RSL records (Bagge et al., 2021). According to their global investigation the 3D viscosity is bounded below by $10^{19}$ Pa s, which keeps integration time manageable for glacial cycle simulations. We do not modify

the 3D Earth structure in view of geodetic inferences as these discuss recent load changes in specific Antarctic regions (e.g.

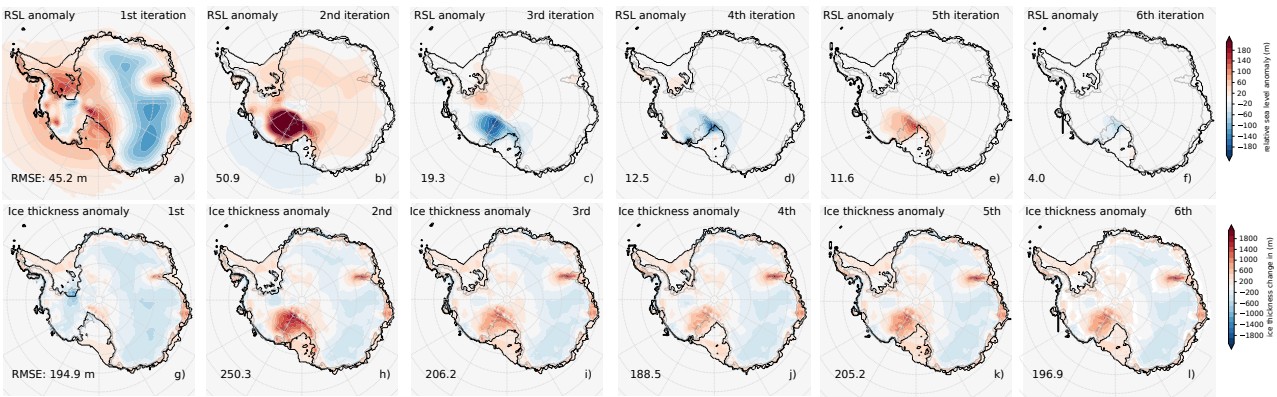

**Figure 4. Anomalies of RSL (VILMA) and ice thickness (PISM) at present-day** compared to Bedmap2 bed elevation and ice thickness observations (Fretwell et al., 2013), for six iterations of coupled glacial cycle simulations with reference 3D Earth structure ('3D ref', see Sect. 2.4), converging from 46 m RMSE to about 4 m RMSE for RSL anomaly (negative bed topography anomaly, cf. black points in Fig. 3c). This iterative procedure does not optimize for the present-day ice thickness. The iterations show a convergence with alternating sign in the mean RSL anomaly, particularly in the Siple Coast region in the Ross Sea sector. Black lines indicate modeled GL and ice shelf extent, grey is the observed GL.

Nield et al., 2014; Barletta et al., 2018), which are not the scope of this study. In the following we refer to the Class-I-type ('v_0.4_s16') as our reference 3D Earth structure ('3D ref').

In order to test for the effects of low and high viscosities on the ice sheet dynamics and to investigate possible feedbacks we define end member Earth structures of the laterally varying reference 3D Earth structure, '3D ref'. The '3D min' and the '3D
max' take the minimum and maximum of '3D ref' over the Antarctic region south of 60°S at each depth interval (see Fig. 5a), respectively. The inferred minimum or maximum define a laterally uniform 1D Earth structure within the Antarctic region, while north of 60°S we consider the original '3D ref' structure, as we want to avoid far-field effects induced by changes in the northern hemisphere Earth structure. This implies a very thin lithosphere thickness of only a few kilometers and upper-mantle viscosities down to $10^{19}$ Pa s in the '3D min' case for Antarctica, while in the '3D max' case, the lithosphere thickness is
around 130 km and upper-mantle viscosities remain larger than $4 \times 10^{20}$ Pa s for all depth intervals (Fig. 5c).

## 3  Results

### 3.1  Sensitivity of AIS deglaciation to spatial resolution and coupling time step

The coupling time step is an important parameter in a coupled model system. As PISM responds to changes in RSL from VILMA of the previous time step, the coupling time step should be short enough to avoid delay effects. However, computational
costs for the remapping of the spatial fields and the initialization of each of the model steps occur at each coupling time step (cf. Table 1). A reasonable coupling time step therefore balances costs and accuracy.

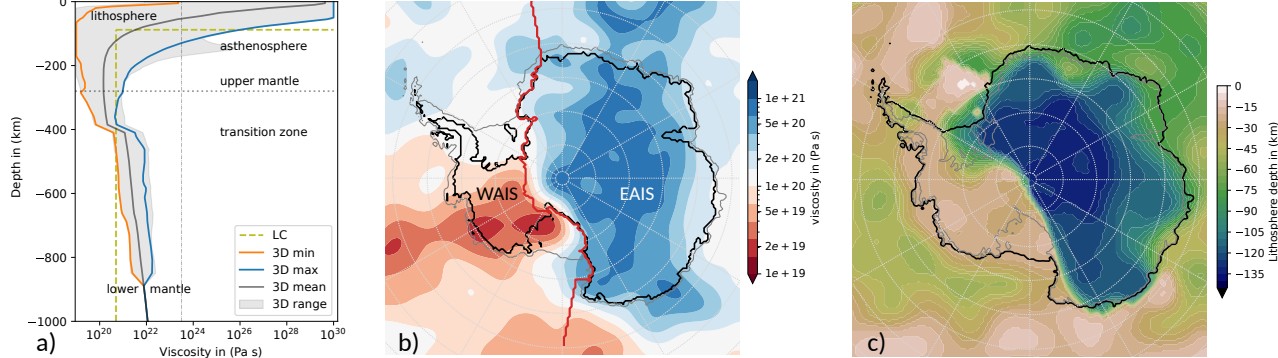

**Figure 5. VILMA Earth structure.** a) Vertical profiles of viscosity down to $1000\,\mathrm{km}$ depth for 3D Earth structures, with global logarithmic mean and range (grey), the Antarctic wide range (between orange and blue defining '3D min' and '3D max', respectively) and the two-layer profile as used in the Lingle-Clark model (olive). b) '3D ref' at $280\,\mathrm{km}$ depth (horizontally dotted line in a), showing a lateral variability of two orders of magnitude ($10^{19}$–$10^{21}\,\mathrm{Pa\,s}$). Present-day grounding line and calving front (Bedmap2 contours from Fretwell et al. (2013), grey and black respectively), and Zwally et al. (2012) drainage basin divide (red) between WAIS (basins 2–17) and East Antarctica (basins 18–27 and 1). **c)** Lithosphere depth defined by $10^{23.5}\,\mathrm{Pa\,s}$ threshold (vertical dashed line in a), showing a thin lithosphere ($<50\,\mathrm{km}$) in West Antarctica and a thick lithosphere exceeding $130\,\mathrm{km}$ in East Antarctica.

We evaluate the sensitivity of the coupled simulations to coupling time steps $\Delta t$ of 1000, 100 and $10\,\mathrm{yr}$ by comparing the deglacial response of the AIS over the last $20\,\mathrm{kyr}$. In all three cases we find a similar SLRP at LGM (in our simulations around $15\,\mathrm{kyr\,BP}$) of around $72.5\,\mathrm{m\,SLE}$, which is about $14\,\mathrm{m\,SLE}$ above the present-day level (see Fig. 6a). At the onset of deglaciation, around $14\,\mathrm{kyr\,BP}$, we find small discontinuities in the response for the largest coupling time step (blue lines), as PISM responds to changes in bed topography that remain constant for the $1000\,\mathrm{yr}$ interval, while the far-field RSL changes dramatically during 'meltwater pulse 1A' (MWP-1a). In fact, this is partly a consequence of the diagnostic calculation of the SLRP, since changes in bed elevation also affect the flotation height and hence potential sea-level contributions. The blue-solid line would be much smoother when total ice mass or grounded ice volume over time were shown. Interestingly, compared to $\Delta t \leq 100\,\mathrm{yr}$, we find during Holocene until present day slightly smaller SLRP (by about $1\,\mathrm{m\,SLE}$) for $\Delta t = 1000\,\mathrm{yr}$. In case of a linear extrapolation of the RSL change rate in the PISM step for this $\Delta t$ (blue-dashed line in Fig. 6a) we find a similar effect during deglaciation, with slightly earlier deglaciation than without extrapolation. Comparing the ice sheet response for $\Delta t = 100\,\mathrm{yr}$ and $10\,\mathrm{yr}$, we see only little differences for the given spatial resolution of n512/n128 in VILMA and $16\,\mathrm{km}$ in PISM, which we interpret as 'convergence'. It should be noted, that the coupled system is highly non-linear with a high sensitivity to initial conditions. Convergence should be considered with respect to a certain range of internal variability (Albrecht et al., 2020a).

As total computational costs per modeled kyr increase by about 20% between $1000\,\mathrm{yr}$ and $100\,\mathrm{yr}$, but by about 65% between $100\,\mathrm{yr}$ and $10\,\mathrm{yr}$, we choose $\Delta t = 100\,\mathrm{yr}$ as the default coupling time step for further experiments.

A recent study on time stepping and RSL precision (Han et al., 2022) with the coupled ice sheet–sea level model system of
Gomez et al. (2013, 2020) suggests coupling time steps of at least 200 yr, while a previous coupled model study with coarser
spatial resolution claimed that 1000 yr would be sufficient (de Boer et al., 2014). Coupling time steps of 500 yr are shown to
yield accurate results, when using internal iterations and linear interpolation over the coupling time step (Van Calcar et al.,
2023). Idealized experiments performed with a coupled model framework with VILMA led to the choice of 50 yr (Konrad
et al., 2015). In fact, an optimal temporal resolution is highly linked to the spatial resolution and the involved Earth structure.

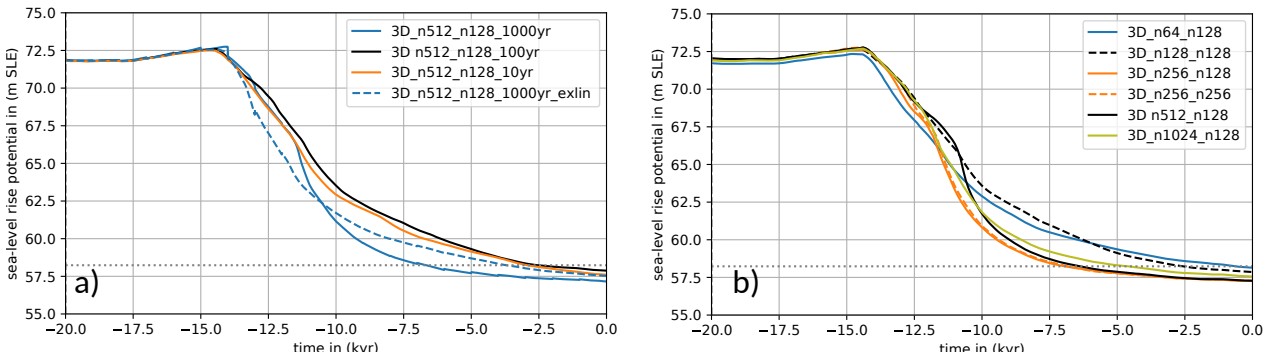

**Figure 6. Sea-level rise potential from Antarctica over last 20 kyr for different coupling time steps and different spatial resolutions.**
a) Default coupling time step is 100 yr (black). Higher temporal resolution with 10 yr is shown in orange. Lower temporal resolution with
1000 yr is shown in blue, and with linar extrapolation ('exlin') of RSL change rate in blue-dashed. Here, all simulations were restarted from
the same restart state at 20 kyr BP ('3D ref', 6th iteration with default spatial resolution). Shown time series have a resolution of 10 yr.
b) Reference spatial resolution is n512 for the sea-level equation in VILMA, n128 for the viscoelastic deformations and 16 km in PISM
(black). Higher spatial resolution for the sea-level equation is shown in olive (n1024), coarser resolution in blue (n64), black-dashed (n128)
and orange (n256). In orange-dashed, also the resolution for the viscoelastic deformations has been increased to n256. All simulations have
been initiated from the same initial state at 246 kyr BP (4th iteration). The coupling time step and temporal resolution of the plotted time
series is 100 yr. PISM resolution has not been varied here.

There are interactive processes between ice sheets, ocean and the solid Earth that need sufficient spatial resolution to be
adequately resolved, in particular in the grounding zones of marine ice sheets. The default spatial resolution in our coupled
simulations is 16 km in the AIS and 0.2° for the global sea-level equation (n512), which corresponds to 20 km in latitude and
to about 6 km in longitude at 71°S. The viscoelastic deformation is resolved with 0.7° (n128), which corresponds to about
78 km in latitude and 25 km in longitude at 71°S. A doubling in the viscoelastic resolution (n256) has only minor effects
on the AIS deglaciation (see orange lines in Fig. 6b). A coarser spatial resolution in the sea-level equation, however, delays
deglaciation during the Holocene (blue-solid and black-dashed lines for n64 and n128, respectively), with a SLRP difference
of up to 2 m SLE compared to the reference resolution. These difference mainly result from RSL anomalies in the Ross Sea
section (Fig. S 1). Only small differences in SLRP are found at around 11 kyr BP for even higher spatial resolutions (cf. olive
and black lines for n1024 and n512, respectively). The resolution of the ice sheet model has not been varied here, as this would

require a new ensemble of simulations to constrain model parameters, as for the complex geometry of the AIS some parameters are resolution dependent (Albrecht et al., 2020a, b).

## 3.2 AIS build-up and deglaciation within bounds of reference 3D Earth structure

The reference 3D Earth structure ('3D ref') discussed in Bagge et al. (2021) implies a large range of lithosphere thickness and mantle viscosity in key regions of the AIS. In order to evaluate effects of lateral variations in the underlying Earth structure on ice dynamics, we discuss changes in the grounded AIS area (GL extent) and its SLRP over the last $246\,\mathrm{kyr}$.

First, we consider the lower bound of '3D ref', i.e. '3D min' (Fig. 5a). In our coupled simulations with '3D min' (3rd iteration), the AIS holds about $14\,\mathrm{m}\,\mathrm{SLE}$ more SLRP at the LGM (here at $15\,\mathrm{kyr}\,\mathrm{BP}$) than observed at present day. This corresponds to a maximum LGM grounded ice area of $15.9$ million $\mathrm{km}^2$ (Fig. 7b) and a total ice mass of $31.2$ million $\mathrm{Gt}$, respectively. For the highly viscous '3D max', the SLRP at LGM reaches only $10\,\mathrm{m}\,\mathrm{SLE}$ above present day, corresponding to $28.9$ million $\mathrm{Gt}$ and $15.1$ million $\mathrm{km}^2$ grounded ice sheet area. In both cases, the WAIS contributes 2–3 times more ice to the rising sea level since LGM than the East Antarctic Ice Sheet (EAIS) (Fig. S 4a). The largest differences in ice sheet response between '3D min' and '3D max' can be detected in the Ross and Ronne Ice Shelf basins and in the Bellinghausen and Amundsen Sea, where for '3D min' locally the RSL is up to $500\,\mathrm{m}$ higher (and hence the underlying bedrock deeper), leading to more extended and more than $1000\,\mathrm{m}$ thicker grounded ice (cf. Figs. 8a, d and Figs. 8b, e, as '3D min' and '3D ref' have similar LGM extent). In Sec. 4, we will discuss this additional feedback mechanism that can explain enhanced GL advance at LGM for weak Earth structures.

In the first iteration, without corrected initial bed topography, we find already similar differences of AIS evolution over time for the different Earth structures, but generally 3–4 m SLE lower SLRP (transparent lines in Fig. 7a, b) and smaller grounded ice sheet areas (transparent lines in Fig. S 2 b) than for the last iteration. During the iterations, bed topography and SLRP converge at present day towards their observed values (Fig. 3c). However, modeled grounding lines at present day tend to slightly expand the observed grounded ice extent (solid and dotted lines in Fig. S 2 b).

The onset of deglaciation in our simulations occurs after MWP-1a around $14.5\,\mathrm{kyr}\,\mathrm{BP}$ (Peltier, 2005), with a global mean sea level rise of about $20\,\mathrm{m}$ during the following $350\,\mathrm{yr}$ (which corresponds to a mean rate of $50\,\mathrm{mm}\,\mathrm{yr}^{-1}$, see Fig. S 4b, c). This far-field effect, mainly in response to the northern hemisphere ice sheet losses (ICE-6G_C), reduces to about $10\,\mathrm{mm}\,\mathrm{yr}^{-1}$ between 14 and $7\,\mathrm{kyr}\,\mathrm{BP}$ (in total around $55\,\mathrm{m}$ of global mean sea level change) and partly compensates for near-field sea level drop in Antarctica, e.g. by viscoelastic bedrock uplift or gravitational attraction (see Figs. 8g, i at $10\,\mathrm{kyr}\,\mathrm{BP}$, cf. Gomez et al. (2020)). During the deglaciation until present-day we find a slightly slower decline, both in SLRP and grounded ice area for '3D min' compared to '3D max' (Fig. 7b and Fig. S 2b). As their SLRP differ by around $4\,\mathrm{m}\,\mathrm{SLE}$ at LGM, they provide different initial conditions at the onset of deglaciation. Grounding line retreat and ice sheet thinning (leading to ice shelf formation) can be slowed down if the isostatic rebound responds to these changes on rather short timescales. For '3D min', RSL change rates are in fact much higher than for '3D max', in West Antarctica reaching up to $-500\,\mathrm{mm}\,\mathrm{yr}^{-1}$ instead of $-20\,\mathrm{mm}\,\mathrm{yr}^{-1}$, while in the far-field, the RSL rises by about $10\,\mathrm{mm}\,\mathrm{yr}^{-1}$ indicated by light-orange shading at the corners of the maps in Figs. 8g, i and Fig. S 4c, d. This quick and rather localized response (e.g. Wan et al., 2022) defines a negative and hence stabilizing feedback

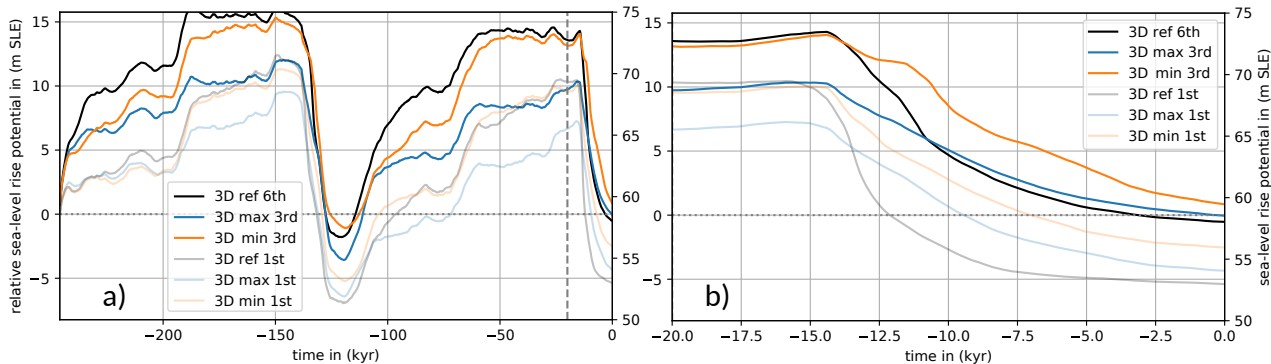

**Figure 7. SLRP from Antarctica over last 246 kyr and over last 20 kyr for different Earth structures.** Blue and orange lines show the response of the AIS in case of the '3D min' and '3D max' (cf. Fig. 5 a), which means respective 1D Earth structures south of 60°S and 3D further north. Black lines indicate the SLRP response to '3D ref', i.e. 3D also at Antarctica. Transparent lines show the first iteration results with same initial topography; dotted horizontal line is present-day observation from Bedmap2 (Fretwell et al., 2013).

on GL retreat (Gomez et al., 2010, 2012). For '3D min', rates of RSLP change and grounded ice area retreat remain almost constant at this comparably low level until present day. The viscoelastic rebound can take several millennia for '3D max', and, in consequence, the stabilizing effect can be significantly delayed and it is active on a larger lateral extent. Over the last 3 kyr the AIS even tends to readvance slightly (Fig. S 2b), as suggested in earlier studies (with PISM-LC) for rather high mantle viscosities of $5 \times 10^{20}$ Pa s (Kingslake et al., 2018).

The reference 3D Earth structure '3D ref' accounts for both, the weak Earth structure in West Antarctica and the higher mantle viscosities and thicker lithosphere in East Antarctica (Fig. 5b, c). The corresponding response of the AIS for '3D ref' shows a similar SLRP (72.5 m SLE) and ice sheet area (15.8 million km$^2$) at LGM as for '3D min' (Fig. 7b and Fig. S 2b, black and orange contour, respectively). The largest differences in the ice loads, compared to '3D max', are located in the West Antarctic region, where a rather low mantle viscosity and a thin lithosphere are dominant (Fig. 8a, d). The additional ice load causes further subsidence of the bedrock underneath, by up to 200 m more than for '3D max' (Fig. S 6), while in the adjacent coastal region the sea floor becomes about 50 m shallower than for '3D max' (Fig. 8a, light blue shading). This feature is called 'forebulge' and results from lateral transport of displaced mantle material and the flexure of the lithosphere. For '3D ref', ice loss rates can reach 1500 Gt yr$^{-1}$ in the first millennia after the onset of deglaciation (Fig. S 4f). This is slightly higher and also earlier than for '3D min' and generally higher than for '3D max' ($< 500$ Gt yr$^{-1}$). Maximum bedrock uplift rates with up to 200 mm yr$^{-1}$ are reached for the low-viscosity response in West Antarctica (Fig. 8h), while maximum uplift rates below 50 mm yr$^{-1}$ are associated with higher viscosities and a thicker lithosphere in East Antarctica (Fig. S 4d and Video S 1).

At present day, the SLRP, grounded ice area and ice mass for '3D ref' (6th iteration) converges against 57.7 m SLE, 13 million km$^2$ and 24.6 million Gt, respectively, which is close to observations (in our diagnostic: 58.2 m SLE, 12.6 million km$^2$, and 24.3 million Gt, dotted lines in Fig. 7a, b and Figs. S 2b, 3b). Counterintuitively, the SLRP response for '3D ref' does not show a trajectory completely in between the two end members '3D max' and '3D min'. Yet, the coupled simulations reveal

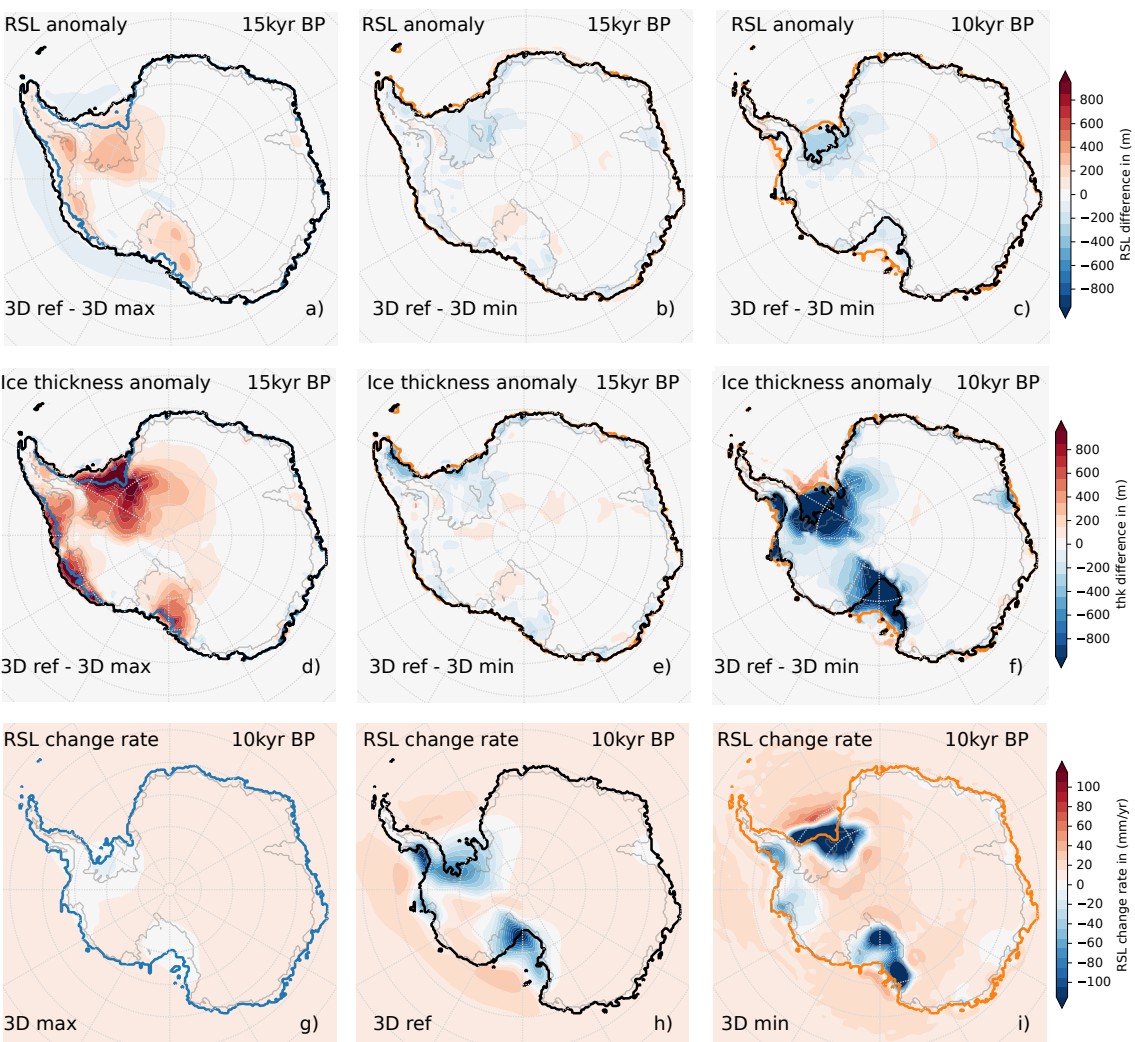

**Figure 8. Difference in relative sea level (a–c) and ice thickness (d–f) for different Earth structures at 15 kyr and 10 kyr BP, and RSL change rates at 10 kyr BP for the different Earth structures (g–i).** The modelled GL position due to the different Earth structures are shown as contour lines for '3D min' (orange), '3D max' (blue), '3D ref' (black) and at present day (grey).

that in glacial periods of GL advance the SLRP trajectory evolves in a similar way as the trajectory for the weak Earth structure ('3D min'), while in deglacial periods of GL retreat the SLRP from the AIS responds in a similar way as the trajectory for the stiffer Earth structure ('3D max'). The similarity of '3D max' and '3D ref' responses during deglaciation is even more pronounced for similar LGM ice sheet conditions (ice sheet history until 20 kyr BP prescribed as in '3D ref', see blue-dashed line in Fig. S 2a).

One important aspect to consider when using 'ice volume above flotation' as metric is that the bed elevation changes imply changes in the grounded ice area and the converted potential sea-level contributions. Other metrics, e.g. the total ice mass, are

independent of bed elevation or sea-level changes. Figure S 3b shows that the differences in total ice mass evolution between '3D ref' and '3D min' are reduced during glacial maxima. Another aspect is a cumulative effect of the performed iterations, with shifted initial bed elevations depending on the present-day anomalies, and hence also on the LGM state and the rate of deglaciation. In the first iteration all simulations are initiated with the same bed elevation and ice thickness distribution, such that differences over time can be purely attributed to effects resultant from the different Earth structures (see transparent lines in Fig. 7a and Figs. S 2b, 3b). With every iteration the differences in ice volumes/masses become larger for different Earth structures, in particular during phases of glacial build-up.

### 3.3 Comparison to glacial-cycle simulations with the PISM-LC model

In order to discuss the impact of the more complete GIA model considered in the PISM-VILMA simulations, we run coupled PISM-LC simulations with the global mean sea level (GMSL) change derived from a coupled simulation with a 1D Earth structure (described in the next paragraph). The resultant GMSL timeseries is similar to the one obtained with '3D ref', as well as to the global mean ICE-6G_C, obtained with 'VM5a' Earth structure (90 km lithosphere and 500 km thick upper mantle with $5 \times 10^{20}$ Pa s), as in all cases the same ice thickness history for the northern hemisphere is used (cf. Fig. 9a and Fig. S 4b). Note that in the PISM-LC coupling the change in GMSL is interpreted as a change in sea surface height. For the same PISM parameters and climatic forcing as in the PISM-VILMA simulations, we find after three iterations with PISM-LC a much lower SLRP from Antarctica, with 59 m SLE at LGM (13.5 m SLE lower than for '3D ref') and 56.5 m SLE at present day (1 m SLE lower than for '3D ref') (olive vs. black in Fig. 9b). Differences in LGM ice volume mainly result from smaller GL extent at LGM, especially in the Weddell Sea sector (Fig. S 6). The PISM-LC results provide an estimate of the influence of GIA, although the inferred LGM ice volume in Antarctica does not seem realistic. Previous model simulations, some of them scored against different paleo records, suggest that the (uncorrected) SLRP at LGM in Antarctica was around 10 m SLE larger than at present day (cf. Fig. S 3a and see Fig. 11b in Albrecht et al., 2020b). For a slightly adjusted PISM parameter combination, a larger glacial ice sheet extent can be reproduced (Kingslake et al., 2018; Albrecht et al., 2020a).

If PISM-VILMA is run, analogous to PISM-LC, with a globally constant viscosity for the whole mantle of $5 \times 10^{20}$ Pa s and a lithosphere thickness of 88 km (see Fig. 5a), we obtain lower SLRP from Antarctica than for '3D ref' (purple vs. black in Fig. 9b). In particular during glacial build-up the SLRP is about 10 m SLE smaller, at LGM around 6.5 m SLE smaller (Fig. 9a, b). This is likely a consequence of the thicker lithosphere and the higher mantle viscosity in large parts of West Antarctica, represented in the simpler Earth model. Compared to the response from the PISM-LC simulations (olive line in Fig. 9b), the SLRP at LGM is about 7 m SLE larger, likely a consequence of potentially stabilizing gravitational and rotational effects accounted for in VILMA. Regardless of the significant differences during glaciated periods, the present-day (and Last Interglacial) SLRP differ by less than 1 m SLE in these simulations (Fig. 9b), as the iterative procedure minimizes the present-day misfit in bed topography.

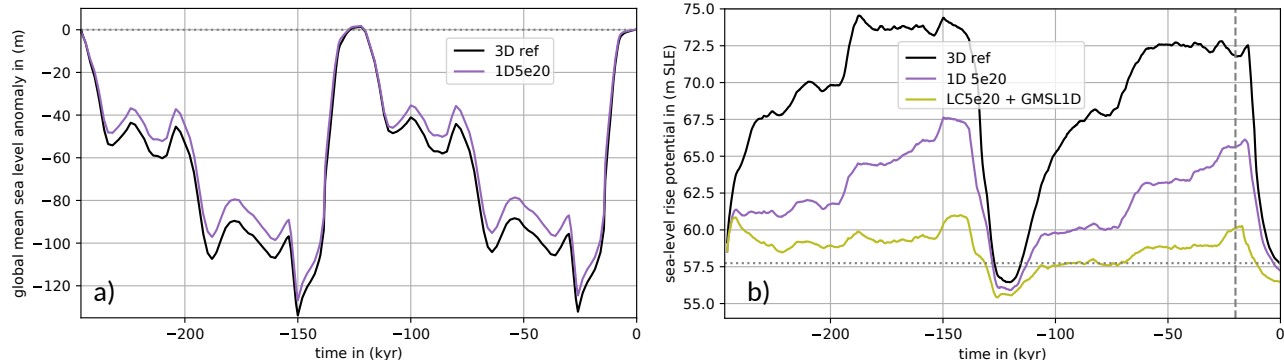

**Figure 9. Comparison of global mean RSL a) and SLRP from Antarctica b) over last 246 kyr for Earth models of different complexity.**
Olive colored for the Lingle-Clark bed deformation model with 100-yr coupling time step, upper mantle viscosity of $5 \times 10^{20}$ Pa s ('LC 5e20')
and 88 km lithosphere thickness ($5 \times 10^{24}$ Nm flexural rigidity), suggesting a loss of about 3 m SLE in SLRP since LGM. Purple for the
coupled PISM-VILMA model with 100-yr coupling time step, a 1D-Earth structure with 88 km thick elastic lithosphere and $5 \times 10^{20}$ Pa s
viscosity in the mantle below ('1D 5e20'). This simulation also solves for the sea-level equation, yielding a 7 m SLE larger contribution since
LGM. The global mean of the sea level is used as forcing for the simulation with the LC model ('GMSL1D'). In black, the AIS response to
the reference 3D Earth structure '3D ref' is shown with more than 14 m SLE change in SLRP since LGM.

### 3.4 Far-field and near-field RSL effects on Antarctic deglaciation

In this section, we focus on the dynamics of the AIS in response to the northern hemisphere ice sheet's decay. We will provide
an estimate of the far-field and near-field components of RSL change since LGM and compare our coupled simulation results
with previous experiments in literature. Gomez et al. (2020) use a coupled ice sheet–GIA model that incorporates a global 1D
(radially varying) viscoelastic mantle structure with four layers: a 50 km thick lithosphere, a 150 km thick low-viscosity zone
of $10^{19}$ Pa s (representative for the weak Earth structure beneath the WAIS), an upper-mantle viscosity of $2 \times 10^{20}$ Pa s and
a lower-mantle viscosity of $3 \times 10^{21}$ Pa s below 670 km depth (see Fig. S 7a). Like in our coupled simulations, Gomez et al.
(2020) prescribe the northern hemisphere ice sheet history (ICE-5G, see anomaly at LGM in Fig. 10a), while the AIS evolu-
tion is simulated with a dynamic ice sheet model (PSU-ISM with 20 km resolution). Sea-level calculations are performed up to
spherical harmonic degree 512, and coupled every 200 yr. Key differences are that PISM-VILMA uses a global 3D Earth struc-
ture and iterations are performed over the last two glacial cycles (246 kyr instead of starting at 40 kyr BP). On the one hand we
have to acknowledge a long memory in the ice sheet's thermal state of the order of 100 kyr (Briggs et al., 2013; Albrecht et al.,
2020a). On the other hand the LGM ice volume depends on the previous glacial build-up and the rate of deglaciation depends
on the LGM ice volume. The potential sea-level contribution from the AIS since LGM (here at 21 kyr BP) is considerably
larger in our simulations (13 m SLE instead of 5 m SLE). In their coupled model setup, the simulated AIS volume changes
since LGM suggest a rather low sensitivity to different radial 1D and 3D Earth structures (Pollard et al., 2017; Gomez et al.,
2018).

In our coupled PISM-VILMA simulations, maximum LGM sea level fall of more than 400 m in the West Antarctic Ross and Weddell embayments is much larger than the 150 m found in Gomez et al. (2020), although the spatial pattern of RSL change is similar. The barystatic mean sea level change since LGM we find at 120 m. If we prescribe the present-day AIS configuration through the GIA simulation, we can estimate the far-field RSL pattern in response to northern hemisphere ice sheet loss since LGM. It has a clear imprint of Earth rotational effects, with 75 m in East Antarctica and up to 125 m in West Antarctica (Fig. 10c), and is in agreement with Gomez et al. (2020). Subtracting the far-field RSL pattern provides a first-order estimate of the near-field GIA effects, mainly the deformational and gravitational GIA components. Accordingly, bedrock elevation in the interior of West Antarctica has been uplifted since LGM by up to 540 m, and in East Antarctica by less than 100 m, see Fig. 10d. At the edge of the continental shelf, a peripheral forebulge of opposite sign developed, with up to 85 m vertical bedrock displacement since LGM. This feature is presumably smaller and hence not visible in the corresponding plots of Gomez et al. (2020), likely as a result of the much smaller deglacial AIS losses.

To test the impact of sea level forcing, we have also performed coupled PISM-VILMA simulations with the northern hemisphere ice history fixed at its stage 40 kyr BP, and hence a continuing far-field RSL of around 100 m below its present day value. We found that the AIS remains almost at its LGM state until present day (Fig. S 7) confirming the importance of sea level in triggering deglacial changes as already stated in Albrecht et al. (2020a). In a similar experiment, Gomez et al. (2020) find a delayed and less pronounced deglacial GL retreat in Antarctica.

## 4 Discussions

A key finding of our coupled ice sheet–solid Earth study is that glacial build-up and deglacial retreat are controlled by different feedbacks, which become dominant for specific viscosity regimes and are associated with respective response time and length scales (see Sect. 3.2). If the GL advances, the upstream grounded ice adds additional load. Subsidence and viscoelastically displaced mantle material produces a forebulge where ice is still floating, which lowers the water depth and enhances GL advance (see schematic loop Fig. 11a). A weaker Earth structure yields faster and hence deeper subsidence in the interior during ice sheet expansion, as well as a more pronounced forebulge uplift. Figures S 5 illustrates the forebulge evolution near the advancing GL, with height and distance depending on the underlying Earth structure, when considering the first iteration and hence same initial bed elevation. A thinner elastic lithosphere allows for more localized response, such that the forebulge induced RSL lowering at the GL can offset the RSL rise due to subsidence and gravitational attraction. GL advance supported by forebulge uplift can reach larger glacial ice sheet extents and hence larger ice sheet volumes, which increases loads and induces further subsidence. This is a positive *forebulge feedback* which stops, when the GL reaches the edge of the continental shelf (Fig. 8e, Figs. S 5a–d).

When temperatures rise and the GL starts to retreat into overdeepened embayments, the Marine Ice Sheet Instability (MISI) supports a self-amplified GL retreat. With a thicker cross section, the GL flux increases non-linearly (Schoof, 2007), which reduces the ice thickness there and leads to further retreat (Fig. 11a). Our coupled simulations suggest that a weaker Earth structure causes larger glacial ice sheet extent and more subsidence of the bedrock. This also implies an even steeper retrograde

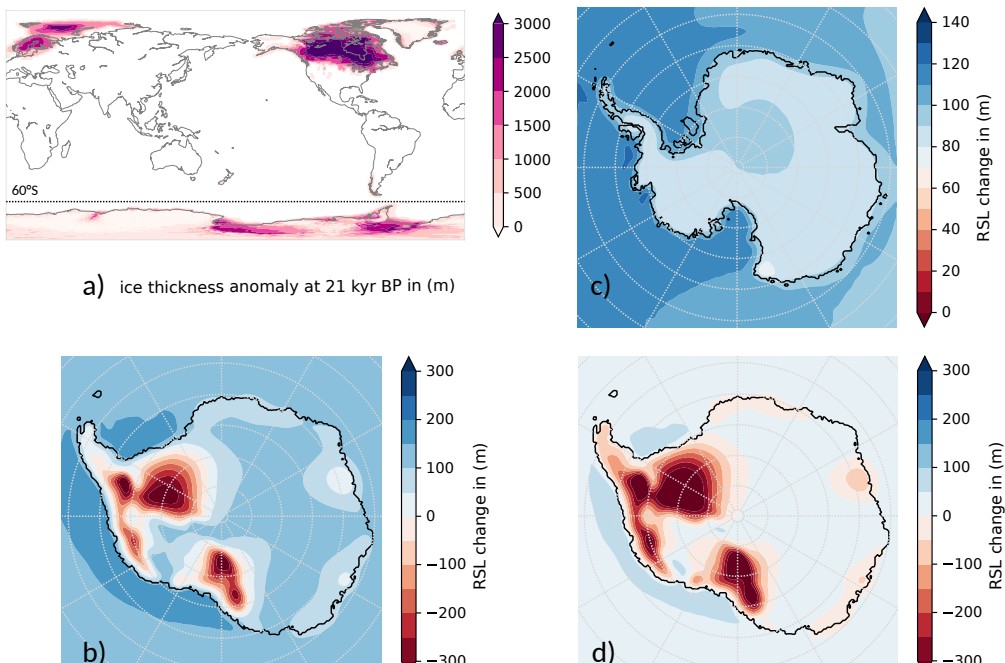

**Figure 10. Far-field and near-field contributions to RSL changes in Antarctica since LGM, analogous to Fig. 1 in Gomez et al. (2020)**
a) Grounded ice thickness anomaly at 21 kyr BP compared to present day in the coupled PISM-VILMA simulation with '3D ref' Earth structure, based on the combination of ICE-6G_C ice history in the northern hemisphere (up to the dashed line at 60°S) and the dynamic ice sheet model for Antarctica. b) RSL change since 21 kyr BP in Antarctica. c) Component of RSL change associated with ice cover changes in the northern hemisphere, computed from a VILMA simulation with constant present-day AIS thickness. d) Difference between b) and c), representing RSL changes associated with Antarctic ice cover changes. These are basically caused by viscoelastic deformation and gravitational effects. Note the difference in color-bar ranges and the values beyond being saturated. The figure design has been chosen similar to Gomez et al. (2020, Fig. 1), to allow a direct comparison.

slope (Fig. 11b) and hence an increased potential for MISI-style GL retreat. However, gravitational changes and viscoelastic
deformations due to GIA dampen the MISI feedback and slow down the retreat. This stabilizing *sea-level feedback* (e.g. Whitehouse et al., 2019) describes how loss in ice mass leads to lowering in RSL and hence to less GL retreat (see Fig. 11a). In our coupled simulations we did not consider the additional effects related to the Marine Ice Cliff Instability (MICI), whose relevance is under debate at the moment (Lipscomb et al., 2024).

Advancing and retreating GLs during Antarctic glacial cycles pass through roughly the same regions. The largest changes in
grounded ice sheet extent (and in RSL) are found in the Weddell Sea and Ross embayments, as well as in the Amundsen and Bellinghausen Sea in West Antarctica, while changes in East Antarctica are comparably small. The weakest Earth structure underneath Antarctica spans between Ross embayment and Amundsen Sea (see Fig. 5). The Earth material becomes stronger towards the Weddell Sea, but is still weaker than the Antarctic-wide logarithmic mean. In the Ross Sea sector the present-day

GL position seems loosely constrained by the geometry of the Siple Coast. This region also shows the weakest Earth structure in our setup with a gradual and delayed modeled GL retreat until present day (Neuhaus et al., 2021). We find here the lowest convergence rates in the iterative procedure (Fig. 4), defined as the relative reduction in RMSE between modeled present-day topography and observation in subsequent iterations, which is in line with the findings by Van Calcar et al. (2023).

For GL advance during the glacial build-up, the Earth structure in all those West Antarctic regions seems to be weak enough to be supported by the forebulge feedback. The maximum LGM grounded ice sheet extent from a coupled simulation with the Antarctic-wide minimum of the 3D Earth structure, '3D min', is similar to '3D ref'. GL retreat during deglaciation tends to occur on much shorter time scales than during glacial build-up. However, due to the sea-level feedback, the retreat can be strongly slowed down in regions with a weak Earth structure, such as in the Ross Sea (Fig. 8). Yet, in the Weddell Sea and Bellinghausen Sea region, the MISI feedback seems to dominate the accelerated GL retreat. The resultant loss in SLRP seems to follow a trajectory, which resembles a coupled simulation with the Antarctic-wide maximum or logarithmic mean of '3D ref', i.e. '3D max' or '3D mean', respectively (see Fig. 7 and Fig. S 2). This implies, that in most Antarctic regions '3D ref' is not weak enough to obtain a significant delay of fast deglacial GL retreat due to the sea-level feedback and seems to be a robust feature, as we find a similar response of the AIS for an even larger variability of 3D Earth structures (Fig. S 8–10).

The strength of the solid-Earth structure can be associated with a characteristic response time scale. In our coupled simulations we find a paradoxical behaviour: A weak Earth structure with comparably short response times tends to support a slow glacial build-up, while it can slow down fast deglacial retreat. As we have a strongly heterogeneous 3D Earth structure, with large differences between East and West Antarctica, but also with moderate differences within West Antarctica, we find a clear asymmetry in the aggregated response to '3D ref': The mantle material is weak enough to support maximum LGM extent, while it is not weak enough to considerably slow down GL retreat. When performing iterations over two glacial cycles, this asymmetric response of the coupled system between glacial and deglacial phases requires larger adjustments in initial bed elevation for '3D ref' than for '3D min', which then supports the glacial build-up in the next iteration and leads consequently to larger glacial ice volumes.

## 5 Conclusions

This study presents a new coupling framework between the ice sheet model PISM and the solid Earth model VILMA. We have run coupled simulations over two glacial cycles with PISM for Antarctica and VILMA with a global 3D viscosity structure. For coupling time steps of $100 \, \mathrm{yr}$ between PISM and VILMA, grid resolutions of $16 \, \mathrm{km}$ for PISM in Antarctica, $0.2°$ ($\leq 20 \, \mathrm{km}$) for solving the sea-level equation and $0.7°$ ($\leq 78 \, \mathrm{km}$) for solving the viscoelastic deformations in VILMA, we are able to capture relevant dynamics and feedbacks in glacial cycle simulations. We performed an iterative correction for initial topography over $246 \, \mathrm{kyr}$ (two glacial cycles) to adjust for the initial bed topography. Much higher resolutions, as may be required for an adequate representation of the complex dynamics at the grounding line (GL) in some key regions, can be applied in this offline coupling framework (e.g. $10 \, \mathrm{yr}$ coupling time step, order of $1 \, \mathrm{km}$ for the ice dynamics and $0.1°$ resolution for the exchange of

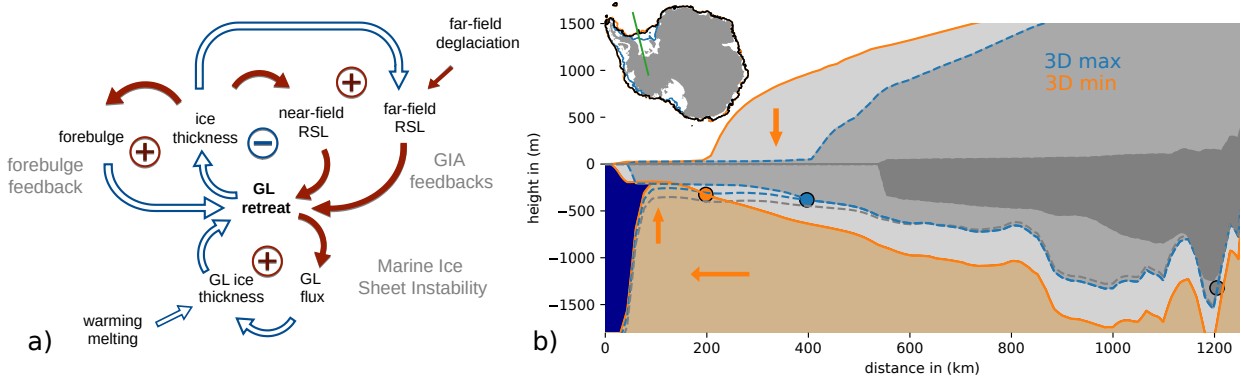

**Figure 11. MISI and GIA feedbacks on grounding line migration.** a) Feedback loops show the stabilizing near-field RSL feedback on grounding line (GL) retreat (-) and the self-sustaining forebulge feedback (+) as well as the far-field (barystatic) RSL feedback (+). The Marine Ice Sheet Instability (MISI) represents a self-amplifying ice-internal feedback on GL retreat on retrograde slopes (+), which can be triggered by ocean-induced melting. Amplifying effects are indicated by red filled arrows (e.g. GL retreat leads to more GL flux), dampening effect by non-filled blue arrows (e.g more GL flux results in less ice thickness at the GL). b) Transect shows bedrock elevation and outlines of ice sheet and ice shelf at LGM through Ronne embayment from the continental shelf edge to the present-day GL position for the two Antarctic end members of the reference 3D Earth structure, '3D max' and '3D min'. Respective GL positions (dots) are about 200 km apart. Orange arrows indicate additional ice mass, subsidence of the underlying bedrock, lateral displacement of mantle material and subsequent forebulge uplift (see also Video S 2). Dark grey shading and dot indicate present-day observed ice sheet configuration and GL position, respectively. Green line in inset indicates transect location.

loading and sea level response); this is possible as the field equations in VILMA are solved in the time-domain, which provides a great flexibility with regard to restart states from which further simulations can be set up.

The tectonic setting of the Antarctic continent reflects strong lateral contrasts in the viscoelastic Earth structure. Our model study highlights the complexity of the interactions between ice sheets, sea level and the solid Earth in Antarctica and world-520 wide, given the spatial variability and uncertainty of the Earth structure and associated characteristic response-time and length scales. We show that competing feedbacks and time scales are at play during Antarctic glacial build-up and deglaciation.

During phases of glacial build-up, when GLs advance over several ten thousands of years, a forebulge can emerge in response to viscoelastic deformations. As the forebulge supports further GL advance, eventually up to the edge of the continental shelf, it defines a self-amplifying feedback on ice sheet growth - which we call the forebulge feedback.

In contrast, during deglaciation, GL retreat can be slowed down as a consequence of the sea-level feedback. In case of a weak Earth structure, it implies that a fast and localized response of the solid Earth can slow down the retreat rate of the ice sheet. On a retrograde sloping bed topography this stabilizing feedback can counteract the self-amplifying Marine Ice Sheet Instability, the latter favoring fast deglaciation. In case of a stronger Earth structure, and hence longer response time scales, we find rather slow and delayed uplift, which can even cause GL re-advance, after it has stabilized at topographic features.

Understanding the interplay of feedback mechanisms and involved time scales is highly relevant for the stability analysis of the AIS in a warming climate. In particular, better constraints on the local Earth structure are required in key sensitive regions of West Antarcica. The GLs of the Amundsen Sea ice shelves (e.g. Pine Island and Thwaites) are likely entry points for the initialization of the West Antarctic Ice Sheet (WAIS) collapse, and also the stability of the Ross Sea and Weddell Sea is under discussion.

Our coupled model system yields self-consistent reconstructions of ice sheet and relative sea level evolutions for 3D solid-Earth structures. The modeled Antarctic sea-level rise potential at Last Glacial Maximum relative to present-day shows a range between 10 and 15 m SLE for the different considered 3D Earth structures. These are slightly larger values than in recent coupled ice sheet–solid Earth reconstructions (Gomez et al., 2020; Van Calcar et al., 2023), but within the range of previous model studies (Albrecht et al., 2020b, Fig. 11b). Based on the gained experience with PISM in high-resolution applications this

study adds confidence for the application of the PISM-VILMA coupling framework to future ice sheet evolution and sea-level projections and the investigation of tipping point characteristics.

*Code and data availability.*  PISM code is freely available and is listed in the Research Software Directory: https://research-software-directory. org/software/pism. VILMA code can be obtained from the authors upon request. The coupling tool is freely available at: https://github.com/ talbrecht/pismvilma. The data and processing code will be made publicly available on a public data repository i.e. PANGAEA or Zenodo.

DOI links to the repositories will be provided upon publication.

*Video supplement.*  Movie of change rate of relative sea level (RSL) and grounding line dynamics in Antarctica over the last 25 kyr from a coupled ice sheet–solid Earth model system, Copernicus Publications: https://doi.org/10.5446/65479, (Albrecht, 2023), Supplement Video S 1. Movie of change of relative sea level (RSL) and grounding line along transect through Ronne embayment in Antarctica over the last glacial build-up phase (123–15 kyr BP from a coupled ice sheet–solid Earth model system for upper and lower bound of 3D reference Earth struc-

ture, Copernicus Publications: https://doi.org/10.5446/68302, (Albrecht, 2024a), Supplement Video S 2.

*Supplement.*  Supplementary material is provided at https://egusphere.copernicus.org/preprints/2023/egusphere-2023-2990/egusphere-2023-2990-suppler pdf

*Author contributions.*  TA designed the study and developed the coupling framework, with great support by MB and VK. MB provided the 3D viscoelastic Earth structures. TA wrote the manuscript and prepared the figures, with contributions from MB. All authors contributed to

the interpretation of model results and revised the manuscript.

*Competing interests.* The authors declare that they have no competing interests.

*Acknowledgements.* TA and MB are funded by the German climate modeling project PalMod (FKZ: 01LP1918A, 01LP1925D, 01LP2305A and 01LP2305B) supported by the German Federal Ministry of Education and Research (BMBF) as a Research for Sustainability initiative (FONA). This work also used resources of the German High-Performance Computing Centre for Climate and Earth System Research (DKRZ) granted by its Scientific Steering Committee (WLA) under project ID bk0993. The authors gratefully acknowledge the European Regional Development Fund (ERDF), the German Federal Ministry of Education and Research and the Land Brandenburg for supporting this project by providing resources on the high performance computer system at the Potsdam Institute for Climate Impact Research (PIK). Development of PISM is supported by NASA grants 20-CRYO2020-0052 and 80NSSC22K0274 and NSF grant OAC-2118285. We thank Reyko Schachtschneider for helping revising the manuscript in an early stage.


*Review statement.* This paper was edited by Alexander Robinson and reviewed by Holly Han and Matt King.

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
