# Peer review of "Feedback mechanisms controlling Antarctic glacial cycle dynamics simulated with a coupled ice sheet—solid Earth model"

_EGUsphere, 2023_

## Referee Comment (RC2)

**Review comments on the submitted manuscript "Feedback mechanisms controlling Antarctic glacial cycle dynamics simulated with a coupled ice sheet–solid Earth model" by Albrecht et al.**

Summary:

Growth or retreat of grounded ice sheets barystatically contributes to global sea level by exchanging mass between cryosphere and hydrosphere. In addition, it changes local relative sea level by causing deformation of the solid Earth and perturbing the Earth's gravitational field and rotation vectors, imprinting spatially varying sea-level changes.

In a little over the past decade, numerous modeling studies have shown the presence and significant impact of the feedback mechanisms between ice sheets, sea level and the solid Earth on ice-sheet modeling, highlighting the importance of coupled ice sheet – solid Earth/sea level modeling, particularly for the marine-based West Antarctic Ice Sheet (WAIS), which is situated upon reverse-sloped bedrock and under threat by MICI, and thus its stability is closely linked to the ocean depth at the grounding line and configuration of bed topography underneath. Moreover, the structure of the Earth across Antarctica is laterally heterogenous; it is characterized by thinner lithosphere and lower mantle viscosity in West Antarctica (WA), which causes faster viscous response of the solid Earth to ice loading/unloading, requiring a rather short coupling interval between the ice sheet model and solid Earth/sea-level model. Thus, capturing the interactions between ice sheets, solid Earth and sea level, as well as considering the 3D structure of the Earth and having a "short" coupling interval in the coupled modeling are warranted for accurate modeling of the Antarctic Ice Sheet.

In their paper, Albrecht and colleagues develop a newly coupled ice sheet-solid Earth/sea level model using the PISM and VILMA, which are a published ice-sheet model and viscoelastic solid Earth model, respectively. They explore Antarctic Ice Sheet dynamics over the last two glacial cycles in simulation experiments with varying Earth structure profiles, number of topographic corrections for the initial topography, coupling interval, and spatial resolution of the solid Earth model. They find a combination of these parameters according to the convergence of the AIS volume configuration and explore sensitivity of AIS to different 3D Earth Structure profiles. In addition to observing the negative sea-level feedback mechanism on grounding line retreat that has been seen by other modeling groups, the authors also observe what they call the "forebulge feedback", which helps the grounding line to advance by reducing the local ocean depth during the glacial buildup phase. The authors also explore and confirm the effect of far-field sea-level forcing from melting of the Northern Hemispheric Ice Sheet during the last deglaciation on the AIS dynamics, which has been previously verified by other studies.

The model development, simulation experiments and analysis done in this manuscript represent lots of hard work by the authors, and undoubtingly an important contribution to the ice sheet and GIA community. It is great to see independent groups around the world building the capability of modeling coupled ice sheet - solid Earth/sea level dynamics; together which will push for better representation of ice-sheet physics and better prediction of future sea level. While I support this topic material is suitable for publication in the journal *The Cryosphere*, I anticipate major revision needs to be done regarding several aspects including more thorough literature review

and correct citation, more exhaustive discussion on outcomes and justifications on experimental choices, which might incur additional simulations.

General comments:

1. It would be helpful to clarify the definition of GIA, whether it solely addresses to the solid Earth deformation component or includes gravitational and rotational effects and use the precise/consistent term for the feedback mechanism. It becomes clear through the text that the authors take the former as the case, but then the term "ice sheet – GIA feedback" becomes imprecise as it misses the changes in the sea surface height.

2. It is great that the authors perform the topographic iteration to have the modeled present-day topography to match with the observed data. However, authors attribute all the differences in results to difference in the Earth Structure and omit the discussion on the role of having different initial bed topography, which could be quite important for ice dynamics. This issue could be addressed by discussing in detail the differences in 3D Earth results given the same initial topography (i.e. 1st iteration) and then putting the 3rd-iteration results into context.

3. The authors observe "forebulge effect" in their simulations, which helps the grounding line to advance during the build up phase. This however seems to warrant more rigorous testing and analyses. (If forebulge effect is real, then I would anticipate the `3D ref` curve should stay below the `3D min` curve in Figure 7a. And maybe that's what you are seeing in the 1$^{st}$-iteration results, but not in the 3$^{rd}$-iteration results, which might suggest the effects related to different initial topography?) You could look at how the local ocean depth changes at the grounding line and few cells from it (or along a transect) through time to check the location and size the forebulge formation. You could also run a simulation on a non-deformable (rigid) Earth for clearer comparison against the deformable Earth cases. Lastly, several other groups (e.g. DeBoer et al. 2014) have seen the destabilizing effect due increase of local water depth at grounding line when ice sheet advances, and putting your results in the context of literature would improve the discussion on this matter.

4. The authors claim that they have performed the convergence test on spatial resolution, along with coupling time step and topographic iteration. However, the spatial resolution has been explored only in the GIA model side, not on the PISM side. Clarifying the wording in the text is advised.

5. There is lack of proper literature review and giving credits to the studies that deserve citation.

6. Despite the high-fidelity of the model and the exhaustive model experiments (e.g. being able to run over glacial cycles and incorporating the 3D Earth Structure), there seems to be a lack of discussion on other phases of the glacial cycles and differences between West and East Antarctica, giving the impression that the discussion is under-utilizing the existing model results.

7. Given that the community is starting to call for clarification and agreement on the method of calculating "ice-sheet contribution to sea level", I recommend authors to provide more detail on how they calculate the sea-level equivalent ice volume: do you correct for all bedrock deformation, density difference between fresh and ocean water, external ocean forcing, or only some of them?

Specific comments:

L3: "In this study, we run coupled ice sheet–solid Earth simulations…"
 "develop" than "run" might be more appropriate.

L36-39: "The redistribution of land ice, ocean water and mantle material induces changes in the Earth's gravity field (with the 'geoid' as equipotential surface), and therefore alters the sea level globally, which follows the geoid (Farrell and Clark, 1976); a correction, which usually is considered as being part of GIA."

This sentence is little awkward. Rephrase? Also, it's missing the direct he Earth's gravity field also gets perturbed by the direct interaction between landice and ocean water (i.e. weakening/strengthening of gravitational attraction) and that the pattern or perturbation is spatially non-uniform.

L39: "Geodetic investigations to… "
What about geophysical evidence? (i.e., seismic studies by Lloyd et al., for example.)

L41: "…a thinner lithosphere, "
The thinner lithosphere in West Antarctica needs to be introduced before referring here.

L34-47:
This paragraph can be further clarified. For now, GIA, geoid change, lateral variations in Antarctica and within West Antarctica are all mentioned here but without clear explanation of the connection between them.

L50: *"Goelzer et al., 2020; Adhikari et al., 2020) "*
These refs don't seem to be the right ones to cite here because 1) these two refs rather talk about a method to calculate ice-sheet contribution to sea level, and 2) the floatation criterion established in a very general sense.

L50: "GIA-induced viscoelastic bed deformation…"
GIA is already introduced as viscoelastic deformation (L36), so the expression "GIA-induced deformation" is double counting the effect. Either use GIA or solid Earth deformation. Also, in this sentence, gravitational effects need to be included to explain the water depth change. As mentioned in the general comment, please clarify the definition of GIA (gravitational and rotational effects included or excluded) and be consistent in using it throughout the text.

L51: "indirectly"
Why is this an indirect effect rather than direct?

L53: "(Gomez et al. 2010, 2012, 2015)"
There are studies that have specifically looked at the effect of gravitational attraction e.g., Coulon et al. 2021; Han et al. 2021 "

L56: "(lapse effect) "
lapse-rate effect

L61: "(Peltier, 2005)"
Is this a right reference for LGM timing? I think it's okay to not have one here unless there's more generic one.

L63: "Such inter-hemispheric "
The inter-hemispheric effect should be first clearly explained before saying "such" effect; the concept is not clear from the sentence before. You can make it clear that the "rapid sea level rise" since the LGM comes from the collapse of the ice sheet from the Northern Hemisphere.

L69: "(Coulson et al., 2021) "
This reference focuses on crustal deformation in response to prescribed ice sheets in the Northern and Southern Hemisphere, so it is doubtful to be an appropriate one to cite in talking about the state of Antarctic Ice Sheet.

L71-83: "Ice sheet models have been coupled to GIA models with different levels of complexity …"
This paragraph should provide more literature review on the coupled ice sheet - GIA models with the highest complexity in the field. For now, it mainly focuses on the models with the lowest level of complexity with ELRA and LC, which do not incorporate gravitational and rotational effects. It would be only fair to add more review on those that already incorporate both gravitational and rotational effects as well as the 3D earth structure.

L71: "(de Boer et al., 2017; Whitehouse, 2018). "
I would recommend citing the studies that have done the actual work of coupling an ice sheet model and a GIA model rather than citing review papers that describe the state of the field. Most of the references that can be cited here are already mentioned elsewhere (i.e. Gomez et al. 2012;2015, deBoer et al. 2017; Konrad et al. 2015, etc.)

L82: "(Goelzer et al., 2020) "
This ref introduces a method on how to correct for sea-level external forcing in calculating ice-sheet contribution to sea level, which happens at the post-processing part, rather than using GMSL as external forcing to drive ISMs?

L83: "with polar motion "
Rotational effects need to be introduced earlier in the introduction section.

L91: "presumably "
doesn't seem necessary.

L94: "Recently, a new generation of GIA models accounts for the 3D Earth structure
(A et al., 2012; van der Wal et al., 2015; Nield et al., 2018; Powell et al., 2021; Blank et al.,
2021) "
The sentence seems incomplete. Also, Latychev et al. (2005) developed a 3D GIA model, and
Powell et al. 2021 used it. And how about the original references for VILMA?

L100-102: "Here, we present a set of new simulations of Antarctic Ice Sheet evolution over the
last 246,000 years (i.e. two full glacial cycles) with the Parallel Ice Sheet Model (PISM) coupled
to the VIscoelastic Lithosphere and MAntle model (VILMA) solving for GIA. "
It would be helpful to describe here in more detail how this study fits into the context of the work
introduced above and in the field. For example, you mention above the interactive coupling
scheme for 1D and 3D ISM-GIA models in the field, what do you similarly or differently? Is
your goal to have a coupled model functionality that can iteratively converge towards an initial
bed topography? Or come up with a coupling timestep that is the most appropriate to use over
the glacial timescale/for a specific load response? What's the reason for doing your experiments
for two glacial cycles?

L115: "interpolated accordingly "
According to what? Does this mean that basal melt is not calculated but rather taken from
somehwere else predefined? Or do you mean the interpolation of basal melt at the ice draft?

L134: "sea level "
"sea-surface height" or "sea surface geoid" would be more precise term to use here.

L149: "rotational deformations "
Do you mean the effect of rotational feedbacks on viscoelastic deformation.

L150: "gravitational "
gravitationally?

L153: "Coast lines can freely migrate according to the local RSL, also accounting for floating ice
shelves "
What does it mean that coastlines migrate accounting for floating ice?

L155: "redistribution of water mass"
"exchange of mass"?

L160:"condition, "

remove since core itself is a boundary rather than a condition?

L160: "we "
is?

L161: "n128"
describe what 'n' is.

L161: "256×512 "
If you are setting the spherical harmonics degree and order to be 170, shouldn't the grid resolution be 160X340?

L164: "gravity-consistent "
Have a consistent expression throughout the text - in Line 150, 'gravitational consistent' is used.

L165: "mass redistribution of water "
Remove "of water".

L168: "…for flexible restarts, the original Fortran77 code was modernized to Fortran90…"
What do you mean by "flexible" restart? Did (or not) VILMA have a restart functionally before but it was restricted with some technical issue? And how was the code modernized? Please be more descriptive.

L169: "calculation of the left-hand side elastic problem and the right-hand side viscous solution "
Don't see any left or right-hand side. Provide an equation or a reference to it?

L169: "For this, openMP4 functionality "
Isn't OPEN MP for parallelization? If so, how does this relate to the restart functionality?

L175:"PISM and VILMA are offline-coupled using a coupling time step that can be rather short. This is an advantage of the explicit time stepping and the weak formulation (in time-domain) of the solid-Earth dynamics (Martinec, 2000) in contrast to the normal-mode approach (in Laplace domain) used by most of the 1D Earth models"
The "coupling time step" in this sentence is the time interval after which PISM and VILMA call the other, and the explicit time stepping referred in this sentence is specifically for VILMA's time stepping. It could be helpful for readers to make connection between the coupling timestep and VILMA's timestep by explicitly mentioning that the coupling interval is constrained by the VILMA time stepping. In addition, the 1D normal mode approach also allows a short time stepping down to annual scale (e.g. Han et al. 2022). So the comparison made in this sentence might not be correct.

Figure 2: "coupling time step "
Include the 'delta t' symbol in the figure.

L179: "…VILMA integrates "
"runs"?

L181: "VILMA interpolates "
What does this mean? Why does ice history need to be interpolated temporally if it's provided by PISM at every coupling interval?

L188: "and water load "
Remove or rephrase unless PISM actually knows about the global water loading distribution.

L190: "the PISM response always lacks behind the VILMA step by one coupling step "
Explain what this means? PISM runs between t0 and t1, and only after then VILMA will run from t0 and t1, and so on; I take you are trying to say that the RSL change from t0 and t1 calculated in VILMA is only reflected in ice dynamics in PISM for t1 and t2, not from t0 and t1.

L213: "(long) memory effects of the AIS to the climate history "
This is not the best reason for running two glacial cycles. Is there a better reason?

Figure 3 caption: "c) Ice volumes above flotation have been calculated considering cell-area weighting and density correction (Goelzer et al. 2020)… "
So you are correcting sea-level change with bedrock change correction term and density correction term from Goelzer et al. 2020? Any external sea-level forcing correction term? Also, include descriptions for the legend in the figure caption.

Figure 3c:
Any idea what's the reason for a big difference in 3D min and max in panel c?

Figure 4.: Is this taking what is labeled as "3D ref" in Fig 3? State it in the caption. Also, please describe the black and grey contour lines.

L235: "previous PISM simulations"
Previous PISM studies?

L247: "The '3D min' and the '3D max'… "
This introduction needs to come when Fig. 3 is first introduced. Also, by looking at the profile in the supplementary figure, these two profiles are laterally homogeneous, and more like 1D Earth profile? If so, would be helpful to state that.

L238 "…viscosity of 5×1020 Pa s). However, this approach is limited to a certain region (e.g. Antarctica) and is unable to solve self- consistently for sea level changes while considering a globally conserved water budget."

This information might be better combined with the part in Introduction where it talks about the limitations of solid earth models with simple structure and no gravitational and rotational effects. It sounds little redundant and out of place here.

261: "sea level equivalent ice volume (in units of mSLE) "
Please describe or provide an equation how this calculation is made. Is this based on VAF without correcting for bedrock elevation change and/or density difference between fresh water and ocean?

L265: "(blue) "
blue lines in Fig. 6

L266: "… 1000 yr delay. "
I might be confused - Why is this considered a 'delay'? Between t1 and t2, PISM is accounting for RSL changes between t0 and t1 calculated by VILMA, so at t1 PISM is not seeing any "delay" in RSL change that happened between t0 and t1?

L266-267:"This is mostly a consequence of the diagnostic calculation of the sea level equivalent ice volume (above flotation) that "
Why would the diagnostic calculation of the SLE VAF influence on the discontinuity, especially just for the 1000yr coupling interval? Even if so, would that be the entire reason? The timing of the appearance of the most pronounced discontinuity seems to coincide with the when Melt Water Pulse 1A event during which so much of the ice sheet melted from the Northern Hemisphere. I wonder if the AIS from PISM experienced a large far-field sea-level feedback from the Northern Hemisphere over each coupling interval, which might have influenced the ice sheet to be more unstable and thus affect the shape of the curve? I'm thinking of the mechanism introduced in Gomez et al. 2020. To check if this really an artifact of the SLE calculation, it would be helpful to show the plot just on ice volume or mass that is a raw output of PISM with no post-processing.

L267: "… also accounts for changes in bed elevation. "
I see. And density correction? Also, provide reference - Goelzer et al. 2020?

L269: "(blue dashed) "
blue-dashed line in Fig. 6

L269-270 & 278: "Comparing the ice sheet response for 100-yr (default) and 10-yr coupling time steps, we find only little differences, which we interpret as 'convergence' "
I think it is important to put the convergence your results within the context of spatial resolution of both PISM and VILMA, as it must depend on both.

L276: "accurate "
precise, rather than accurate.

Figure 6a:

What is 'dbdt' in blue dashed line in subplot a)?

L285: "blue and black dashed lines"
blue solid and black dashed lines

L284-286: "A coarser spatial resolution in the sea level equation, however, delays deglaciation during the Holocene (blue and black dashed lines for n64 and n128, respectively), with up to 2m SLE difference to the reference resolution. "
Any idea why would it be this way?

Figure 6:
The black solid line in panel b) and a) should be equivalent based on the description (3D n512_n128 & dt 100 yr), but they don't seem to be. Also the black solid line in panel b) shows some weird curvature around 11kyr BP, jumping away from convergence. Is this why is this happening? As it is no clear pattern of convergence towards the black solid line in Fig. 6b, I wonder what's the rational for the authors decision to take this run (3D n512 n128) to expand onto doing sensitivity test to coupling interval.  I suggest authors to re-check their plots (including the consistency between the black solid lines in Fig. 6a and Fig.6b) and re-do the experiment if necessary.

L290: "ice-dynamical effects for variation of the underlying Earth structure, "
You mean to say "effects of lateral variations in the underlying Earth structure on ice dynamics"?

L295: "mio. "
Describe what this is.

L313: "e.g. by viscoelastic bedrock uplift"
Include the other effects too since sea level change happens combination of changes in the sea surface height and bedrock height.

L319: "light orange shading, "
the orange shading is not just far-field effects but combination with near-field effects, but it sounds like the light orange shading only represents far-field effects.

L320: "cf. Wan et al., 2022 "
Wan et al. explores only the GIA model, not coupled ISM-GIA model. The original coupled modeling studies that saw this effect would be the ones to cite here. Also, make the use of 'cf' consistent throughout the text.

L321:"rates of ice volume change and grounding line retreat remain comparably high until present "
But it's mentioned earlier the ice volume (SLE) has slower decline during the deglaciation phase.

L341:"'3D ref' shows a trajectory in between those two end members "

This does not seem to be true for the post-LGM deglaciation phase, and the reason need to be discussed clearly in the text.

L343:"characteristics due to the weak Earth structure ('3D min') seem to dominate the overall ice sheet response, … "
Describe what this means.

L366: "Antarctica"
Represented in the simpler Earth model.

L366: "which also alters the global mean sea level "
Little confusing what "which" refers to and what this sentence means as a whole, regardless.

L365: "7mSLE"
This seems rather a huge difference (around 10% of the total LGM SLE). The difference might be reduced if the 1D model incorporated a lower upper mantle viscosity (($10^{18}$ or $10^{19}$ PaS), which is a better approximation of the viscosity in the West Antarctic region.

L372: "In our coupled simulations we focus on the dynamics of the AIS in response to the northern hemisphere ice sheet's decay.
Add "For this section,".

L376"weak structure "
weak Earth structure

L381: "a synchronous coupling. "
If you mean "synchronous" as the ISM and GIA model communicate every coupling time step, Gomez et al. also uses synchronous coupling. Also, the coupling scheme presented in this work doesn't seem to be different in Gomez et al.'s.

L381: "we run iterations over the last two glacial cycles (246 kyr instead of 40 kyr). "
Given that the LGM deglaciation signal would be so dominant for the deglacial period (post-LGM), how might running over two glacial cycles improve your results compared to running one glacial cycle from the LGM peak?

L389: "As barystatic mean sea level change since LGM we find 120m (about 107m from the northern hemisphere in ICE-6G_C) "
It's already mentioned in Line 383 that 13m SLE is seen from your simulation.

L390: "we prescribe the present-day AIS configuration, "
do you mean when you "fix" the AIS through the GIA simulation and just see the fingerprint of the NH ICE6GC on Antarctica?

L378: "ICE-5G, "
Gomez et al. 2020 also did sensitivity test using different NH ice histories using ICE6GC.

L393: "viscoelastic "
viscoelastic deformational

Figure 10:
The figure captions are almost identical to the one from Gomez et al. I don't know the rule for when you are reproducing plots from other papers, but you should make sure you are not plagiarizing by accident.

Figure10:"analogous to Gomez et al. (2020) "
Include the figure number from Gomez et al. as well?

L395: "forebulge effect "
The last time this term appears is in the abstract, and it would be helpful to have a clear description on what this is before using this term.

L396: "This effect is presumably smaller "
Possibly because the ice mass loss in Gomez et al. is much smaller than the simulation shown in this work. (referring to the comparison made in L383).

Figure 8:
Describe the contour lines

L428: "present-day"
Remove "-"

L428: "lowest convergence rates "
you mean the "lowest number of iteration" required to converge modeled present-day topography to observation?

L430: "The maximum LGM extent is very similar to the ... "
Inferred from what?

L432: "then "
Than

L433:"GIA feecback"
sea-level feedback seems to be a more precise word to use because the definition of GIA in this work seems to only represent the deformational part.

L450:"…as sufficient for capturing the relevant dynamics and feedbacks in glacial cycle simulations."

Sufficient seems to be quite a vague word and maybe misleading. While it's true you are able to see differences in coupling vs. no-coupling with the coarse resolutions, it would be important to acknowledge the importance of high-resolution for capturing accurate ice dynamics at the grounding lines shown in many modeling studies.

L457: "higher resolutions are possible "
Is this true for the ice model as well?

L464: "slow response "
More appropriate expression would be something like "slows the retreat rate of the ice sheet."

L467: "which can even cause grounding line re-advance. "\
But shouldn't this be the other way around because on slow uplift on retrograde-sloped bed would leave the MISI to keep acting.

L451:"outer iteration "
This word has not been used in the text. I would suggest writing "topographic correction" or "iterative correction for initial topography" or some sort that is much more clear.

L477: "This provides confidence in the application of the PISM-VILMA coupling framework to future ice sheet and sea level projections and the investigation of tipping point characteristics "
This is a rather strong statement to make. Ice-sheet models that focus on predicting future sea level emphasize the importance of high-resolution and higher-order velocity solvers for capturing rapid evolution at grounding lines as well as small-scale peripheral glaciers. Plus, the work by Albrecht et al. 2020b. uses the same ice-sheet model, PISM, so I'm not sold by this argument. As mentioned in a comment above, the importance of high-resolution ice sheet modeling for grounding line dynamics and the short-coming of the model used here need to be acknowledged.
* * *
References:

Coulon et al 2021. Contrasting Response of West and East Antarctic Ice Sheets to Glacial Isostatic Adjustment

Latychev et al. 2005. Glacial isostatic adjustment on 3-D Earth models: a finite-volume formulation

Han et al. 2021. Modeling Northern Hemispheric Ice Sheet Dynamics, Sea Level Change, and Solid Earth Deformation Through the Last Glacial Cycle

---

## Author Comment (AC2)

**Reply to RC2: Holly Han, 29 Feb 2024**

Citation: https://doi.org/10.5194/egusphere-2023-2990-RC2

Review comments on the submitted manuscript "Feedback mechanisms controlling Antarctic glacial cycle dynamics simulated with a coupled ice sheet–solid Earth model" by Albrecht et al.

Summary: Growth or retreat of grounded ice sheets barystatically contributes to global sea level by exchanging mass between cryosphere and hydrosphere. In addition, it changes local relative sea level by causing deformation of the solid Earth and perturbing the Earth's gravitational field and rotation vectors, imprinting spatially varying sea-level changes.
In a little over the past decade, numerous modeling studies have shown the presence and significant impact of the feedback mechanisms between ice sheets, sea level and the solid Earth on ice-sheet modeling, highlighting the importance of coupled ice sheet – solid Earth/sea level modeling, particularly for the marine-based West Antarctic Ice Sheet (WAIS), which is situated upon reverse-sloped bedrock and under threat by MICI, and thus its stability is closely linked to the ocean depth at the grounding line and configuration of bed topography underneath. Moreover, the structure of the Earth across Antarctica is laterally heterogenous; it is characterized by thinner lithosphere and lower mantle viscosity in West Antarctica (WA), which causes faster viscous response of the solid Earth to ice loading/unloading, requiring a rather short coupling interval between the ice sheet model and solid Earth/sea-level model. Thus, capturing the interactions between ice sheets, solid Earth and sea level, as well as considering the 3D structure of the Earth and having a "short" coupling interval in the coupled modeling are warranted for accurate modeling of the Antarctic Ice Sheet.

In their paper, Albrecht and colleagues develop a newly coupled ice sheet-solid Earth/sea level model using the PISM and VILMA, which are a published ice-sheet model and viscoelastic solid Earth model, respectively. They explore Antarctic Ice Sheet dynamics over the last two glacial cycles in simulation experiments with varying Earth structure profiles, number of topographic corrections for the initial topography, coupling interval, and spatial resolution of the solid Earth model. They find a combination of these parameters according to the convergence of the AIS volume configuration and explore sensitivity of AIS to different 3D Earth Structure profiles. In addition to observing the negative sea-level feedback mechanism on grounding line retreat that has been seen by other modeling groups, the authors also observe what they call the "forebulge feedback", which helps the grounding line to advance by reducing the local ocean depth during the glacial buildup phase. The authors also explore and confirm the effect of far-field sea-level forcing from melting of the Northern Hemispheric Ice Sheet during the last deglaciation on the AIS dynamics, which has been previously verified by other studies.

The model development, simulation experiments and analysis done in this manuscript represent lots of hard work by the authors, and undoubtingly an important contribution to the ice sheet and GIA community. It is great to see independent groups around the world building the capability of modeling coupled ice sheet - solid Earth/sea level dynamics; together which will push for better representation of ice-sheet physics and better prediction of future sea level. While I support this topic material is suitable

for publication in the journal The Cryosphere, I anticipate major revision needs to be done regarding several aspects including more thorough literature review and correct citation, more exhaustive discussion on outcomes and justifications on experimental choices, which might incur additional simulations.

**This is a great summary of our work and we are very happy about the reviewer's very positive assessment. We will try our very best to cover all mentioned aspects in the revised manuscript. We did perform many more simulations and sensitivity tests, but decided at the end to the presented subset. Find below our point-to-point response and comments in bold black font.**

General comments:

1. It would be helpful to clarify the definition of GIA, whether it solely addresses to the solid Earth deformation component or includes gravitational and rotational effects and use the precise/consistent term for the feedback mechanism. It becomes clear through the text that the authors take the former as the case, but then the term "ice sheet – GIA feedback" becomes imprecise as it misses the changes in the sea surface height.

**We agree that terminology is an important aspect. GIA describes the gravitationally consistent deformational response of the Earth to loading processes related to glacial surface processes. Accordingly, the ice sheet—GIA feedback can be considered as part of the GIA process. Our understanding of GIA also encompasses changes in sea-surface height (GRD). In L36ff we already mentioned that: 'The redistribution of land ice, ocean water and mantle material induces changes in the Earth's gravity field (with the 'geoid' as equipotential surface), and therefore alters the sea level globally, which follows the geoid (Farrell and Clark, 1976); a correction, which usually is considered as being part of GIA'. To clarify this we will moderately alter the paragraph: 'This gravitational-rotational-deformational (GRD) correction is usually considered as being part of GIA.'**

2. It is great that the authors perform the topographic iteration to have the modeled present-day topography to match with the observed data. However, authors attribute all the differences in results to difference in the Earth Structure and omit the discussion on the role of having different initial bed topography, which could be quite important for ice dynamics. This issue could be addressed by discussing in detail the differences in 3D Earth results given the same initial topography (i.e. 1st iteration) and then putting the 3rd iteration results into context.

**This is certainly an interesting aspect, which we have not explicitly addressed. The initial ice thickness and bed topography in the first iteration is identical in all simulations, and hence also the sea-level relevant ice volume above flotation (about 58.0 m SLE). In the last iterations, the corrected initial bedrock topography is more than 300 m higher in parts of WAIS (*Fig. AC2*).**

[Figure]

[Figure]

[Figure]

*Fig. AC2: Difference in initial topography relative to Bedmap2 present-day observations in the 6th (3Dref, left) iteration or 3rd iteration (3Dmax in the middle, 3Dmin to the right), showing much higher initial bed elevation in WAIS, but lower elevation in EAIS. Compared to the 3Dref case, the correction in the Weddell Sea sector is less pronounced in the 3Dmax case, and even less in the 3Dmin case.*

As the used conversion based on ice volume above flotation (VAF) takes into account changes in bed topography (for the same ice masses), this results in different initial sea-level relevant ice volumes of 59.4, 58.7 and 58.9 m SLE for 3Dref, 3Dmax and 3Dmin, respectively *(see Fig. AC3a).* Hence, at 246 kyr BP about 0.5 m SLE between 3Dref and 3Dmin is explained only by the used diagnostic for different initial bed topography. And this aspect is also relevant during glaciation, where bedrock subsidence is fastest and most pronounced in the 3Dmin case.

When considering aggregated ice mass (in million Gt), the differences between 3Dref and 3Dmin (black and orange) are much smaller (*Fig. AC3b*). In the first iteration, ice mass changes can be attributed purely to the dynamical effects of the coupled model for the different Earth structures. The ice mass evolution for 3Dref and 3Dmin are quite similar during glaciation but differ during deglaciation (due to the GIA-sea level feedback). The difference between 3Dref and 3Dmax (black and blue) are most pronounced during glaciation.

The initial bed topography of the second iterations (and each further iteration) is determined by the rate of deglaciation and hence by the offset of the RSL at present (at the end of the previous iteration). The different initial topographies have indeed an effect during build-up of the Antarctic Ice Sheet, as the reviewer mentioned correctly, generally leading to more ice build-up. We will discuss these aspects in more detail in the revised manuscript.

[Figure]

*Fig. AC3: Sea-level relevant ice volume, analogous to Fig. 7a in manuscript, but for the full 246 kyr time span, as well as ice mass over two glacial cycles, for three Earth structures, in first and last iterations.*

3. The authors observe "forebulge effect" in their simulations, which helps the grounding line to advance during the build up phase. This however seems to warrant more rigorous testing and analyses. (If forebulge effect is real, then I would anticipate the `3D ref` curve should stay below the `3D min` curve in Figure 7a. And maybe that's what you are seeing in the 1st-iteration results, but not in the 3rd-iteration results, which might suggest the effects related to different initial topography?) You could look at how the local ocean depth changes at the grounding line and few cells from it (or along a transect) through time to check the location and size the forebulge formation. You could also run a simulation on a non-deformable (rigid) Earth for clearer comparison against the deformable Earth cases. Lastly, several other groups (e.g. DeBoer et al. 2014) have seen the destabilizing effect due increase of local water depth at grounding line when ice sheet advances, and putting your results in the context of literature would improve the discussion on this matter.

**The reviewer raised some great questions to consider. As discussed above, the 3Dref ice mass (or sea-level relevant ice volume) is in fact very close to the 3Dmin response in the first iteration, or even below it (*Fig. AC3*).**
**For the same initial bedrock elevation in the first iteration, we find a clear signal of a peripheral forebulge in the RSL anomaly during glaciation (246–150 kyr BP, 120--15 kyr BP, *Fig. AC4*).**

[Figure]

[Figure]

[Figure]

[Figure]

*Fig. AC4: Difference in RSL between 3Dref and 3Dmax in first iteration at four snapshots during last glaciation, showing peripheral forebulge (red shading) near advancing grounding lines (black for 3Dref and blue for 3Dmax). Grey line indicates transect location.*

For low mantle viscosities (3Dmin and 3Dref), a forbulge within a distance of 100-200 km forms with a relative sea level below the global mean sea level, while for higher mantle viscosities (3Dmax or purely elastic) the forebulge does not form at all (or has a much larger distance from the grounding line, *see Fig. AC5*). The localized influence of the forbulge affects the RSL at the grounding line position and counteracts the sea level rise due to subsidence and gravitational attraction of the additional ice masses. Hence, we hypothesize that the forbulge in case of low mantle viscosities can support grounding line advance on glacial time scales.

[Figure]

*Fig. AC5: Bedrock change along transect in Ronne Ice Shelf (analogous to Fig. 11b in manuscript, see grey line in Fig. AC4d) with respect to initial bed topography in first iteration (dotted) at 25 kyr BP, at the end of the last glaciation, centered around the actual grounding line position (dashed). Subsidence in the grounded part to the right leads to uplift of a peripheral forebulge in front (left) of the advancing grounding line. Positive values mean a water depth (RSL) lowering, which is a combination of viscoelastic, but also gravitational and rotational effects on top of the barystatic (mean) sea level change (dash-dotted).*

Testing for the rigid (non-deformable) case is difficult in the VILMA setting, as the viscoelastic deformational part can not be isolated from the barystatic, gravitational and rotational effects on the relative sea level (geoid). When we compare a rigid PISM standalone simulation (no viscoelastic bed deformation considered), forced with the global mean sea level, with the coupled simulations with 3Dref Earth structure, we find larger sea-level relevant ice volumes (but slightly smaller ice masses) at LGM in the first iteration (*Fig. AC6, purple*). As we find in the coupled simulations generally higher relative sea-levels at the GL than in the far-field mean (cf. *Fig. AC5*), one could expect larger glacial grounding line extent than in the coupled model simulations (in line with previous literature, see below). However, for low mantle viscosities the viscous forebulge counteracts this effect, such that

glacial build-up compares with the result from the standalone ice sheet model case. In a purely elastic coupled case (extreme high viscosity), we find ice volumes close to the 3Dmax case (*Fig. AC6, brown*), revealing the contributions of viscous processes in Antarctic Ice Sheet glaciation.

[Figure]

[Figure]

*Fig. AC6: Sea-level relevant ice volumes above flotation and grounded ice volumes over two last glacial cycles in first iterations (same initial ice thickness and bed topography) for the 3Dref, 3Dmin and 3Dmax case. For comparison, also results in the elastic limit are shown as well as in a rigid uncoupled ice sheet simulation, forced with 3Dref global mean sea level change.*

In the literature, peripheral forebulges are described as a broad-scale viscous process that results from bending of the elastic lithosphere the return flow of mantle material assuming viscous incompressibility (Adhikari et al., 2014; Wan et al., 2022). During deglacial or interglacial periods, the forebulge subsidence is often associated with a mechanism termed ocean syphoning (Mitrovica and Milne, 2002; Mitrovica and Peltier, 1991), which draws water from the far field and thereby contributes a far-field sea-level fall (Powell et al., 2021; Yousefi et al., 2022).

Coupled model studies suggest that in the interaction of GIA and grounding line advance in cooling (glaciation) phases, smaller glacial extents (and slower GL advance and smaller ice volume) can be expected than in uncoupled simulation (forced with global mean sea level), as self-gravitational pull effects and local RSL changes, which can be associated with the negative (self-stabilizing) effect of the RSL (Gomez et al. 2013; DeBoer et al., 2014, 2017). This is in line with our findings described above.

Konrad et al., 2014 find substantial differences between coupled simulations with VILMA and a simple ELRA model, mainly in the region of peripheral forebulge, which is relevant for interpreting constraints on GIA in the periphery of the ice sheet, such as sea-level indicators and GPS uplift rates.

The latitudinal extent of the forbulge region strongly depends on the lithosphere thickness. It can be either confined close to the coast and inner continental shelf for a thin lithosphere or reach offshore for thick lithospheres (Stocchi et al., 2013). Local crust motion and slopes may changes sign during lateral motion of the topographic bulge, which makes it "extremely difficult to isolate and quantify, and it is therefore not obvious whether (and under what circumstances) each of these acts to accelerate or inhibit the ice flow" (Adhikari et al., 2014).

We will add more details of this extended literature review on the forebulge effect to the revised manuscript (see also comment 5 below). However, we have not found any description of the forebulge feedback as we discuss it in our study.

4. The authors claim that they have performed the convergence test on spatial resolution, along with coupling time step and topographic iteration. However, the spatial resolution has been explored only in the GIA model side, not on the PISM side. Clarifying the wording in the text is advised.

**Yes, increasing the resolution for PISM (<16km) would require a new ensemble of simulations to constrain model parameters, as some are resolution dependent. Convergence tests for increasing resolution in ice sheet models applied to complex geometries is hence not trivial. We will scrutinize the text regarding this aspect.**

5. There is lack of proper literature review and giving credits to the studies that deserve citation.

 **We will add some of these previous works mentioned above to the revised manuscript.**

6. Despite the high-fidelity of the model and the exhaustive model experiments (e.g. being able to run over glacial cycles and incorporating the 3D Earth Structure), there seems to be a lack of discussion on other phases of the glacial cycles and differences between West and East Antarctica, giving the impression that the discussion is under-utilizing the existing model results.

**The manuscript is already long and covers many interesting aspects. In terms of investigated feedbacks we roughly focussed in this study on phases of glaciation (GL advance on long time scales) and deglaciation (GL retreat on shorter time scales) during the glacial cycles. We have more confidence in the last glacial cycle results than in the penultimate glacial cycle result, where we had to make assumptions for the northern Hemisphere ice history (similar to Han et al., 2022, Sect. 3.3.1). Most of these feedbacks reveal their dynamics mainly in the marine portions of West Antarctica. In the revised manuscript we will elaborate more on how the dynamics in the different phases affect each other and discuss consequences for subsequent iterations, and we may have a closer look to the marine sections in East Antarctica.**

7. Given that the community is starting to call for clarification and agreement on the method of calculating "ice-sheet contribution to sea level", I recommend authors to provide more detail on how they calculate the sea-level equivalent ice volume: do you correct for all bedrock deformation, density difference between fresh and ocean water, external ocean forcing, or only some of them?

**We agree, that the used conversion method is important to state. We will refer in the revised manuscript to the definition provided in Albrecht et al., 2020a. We use a cell-area weighted and density-corrected version of the volume-above-flotation approach (VAF) to convert into units of m SLE. This makes our numbers comparable with the literature (where VAF is often used), here in the notation by Goelzer et al., 2020:**

$$SLE_{af} = \frac{\rho_{ice}}{\rho_{water}A_{ocean}} \sum_{n} \left( max \left[ H_n + min(b_n - z_0, 0)\frac{\rho_{ocean}}{\rho_{ice}}, 0 \right] \right) \frac{A_n}{k_n^2}$$

**No external sea-level forcing (z_0) is used, the total RSL change from VILMA is interpreted by the ice sheet model (PISM) as negative bed elevation change (b_n), as it corresponds to the geoid change. The sea-level contribution is estimated on the regional PISM grid and it still bears a signal of bedrock changes (as highlighted above). Adhikari et al., 2020 corrects for the contribution of regions grounded below sea level, relative to present (*see Fig. AC7, purple*). Goelzer et al., 2020 expands the correction also into ice-free regions, accounting for water expulsion effects (*brown*). In our RSL data, which is a combination of the deformational, gravitational but also the rotational effectss, the far-field sea level change with distance from the AIS explains most of the gap between LGM sea-level contributions (*Fig. AC7*).**

[Figure]

*Fig. AC7: Global mean sea level change over last 20kyr from coupled simulations with 3Dref Earth structure, estimated from Voluma above flotation (VAF), with density correction as in PISM and with further corrections in marine sections as in Adhikari et al., 2020 and Goelzer et al., 2020.*

Specific comments:

L3: "In this study, we run coupled ice sheet–solid Earth simulations…"
"develop" than "run" might be more appropriate.
**"We developed a coupling scheme and performed a suite of coupled ice sheet–solid Earth simulations…"**

L36-39: "The redistribution of land ice, ocean water and mantle material induces changes in the Earth's gravity field (with the 'geoid' as equipotential surface), and therefore alters the sea level globally, which

follows the geoid (Farrell and Clark, 1976); a correction, which usually is considered as being part of GIA."

This sentence is little awkward. Rephrase? Also, it's missing the direct he Earth's gravity field also gets perturbed by the direct interaction between landice and ocean water (i.e. weakening/strengthening of gravitational attraction) and that the pattern or perturbation is spatially non-uniform. **The direct perturbation between land ice and ocean water is meant implicitly. Maybe the term 'globally' was a bit misleading to the reviewer. We write now as following:"The consistent redistribution of land ice, ocean water and mantle material implies an alteration of the gravitational potential to which in turn the sea level adjusts (Farrell and Clark, 1976). We consider this as being part of the GIA process."**

L39: "Geodetic investigations to…
" What about geophysical evidence? (i.e., seismic studies by Lloyd et al., for example.)
**We will clarify this in conjunction within the paragraph "L34-47". Lloyd et al., 2020 has been cited in L44 with regard to the dichotomy in Earth structure between West and East Antarctica. We added to L39: "Geodetic and geophysical investigations of crustal uplift in the Amundsen Sea sector in West Antarctica (Barletta et al., 2018; Lloyd et al., 2020; Blank et al., 2021) postulate a laterally strongly varying viscoelastic Earth structure with mantle viscosities orders of magnitude lower than the global average."**

L41: "…a thinner lithosphere, "
The thinner lithosphere in West Antarctica needs to be introduced before referring here.
**We added "and in the likely case of an also thinner lithosphere" .**

L34-47:
This paragraph can be further clarified. For now, GIA, geoid change, lateral variations in Antarctica and within West Antarctica are all mentioned here but without clear explanation of the connection between them.
**We agree that the geophysical techniques are not specified here, we delete some of the references and rephrase: "Furthermore, geophysical investigations demand a clear dichotomy (Behrendt, 1999; Morelli and Danesi, 2004; Accardo et al., 2014; ; ; Lloyd et al., 2020; Powell et al., 2020),"**

 L50: "Goelzer et al., 2020; Adhikari et al., 2020) " These refs don't seem to be the right ones to cite here because 1) these two refs rather talk about a method to calculate ice-sheet contribution to sea level, and 2) the floatation criterion established in a very general sense.
**That's right, but both methods are based on the `volume above flotation' approach with some corrections. As we introduce the RSL in this sentence, we will add the sentence: "... directly determines the grounding line position of marine ice sheets via the flotation criterion. A global mean (barystatic) sea level can be calculated from the global RSL change, or by aggregating changes in ice sheet**

**thickness and near-field RSL changes, based on the `volume above flotation' (Gregory et al., 2019; Goelzer et al., 2020; Adhikari et al., 2020)."**

L50: "GIA-induced viscoelastic bed deformation…"
GIA is already introduced as viscoelastic deformation (L36), so the expression "GIA-induced deformation" is double counting the effect. Either use GIA or solid Earth deformation. Also, in this sentence, gravitational effects need to be included to explain the water depth change. As mentioned in the general comment, please clarify the definition of GIA (gravitational and rotational effects included or excluded) and be consistent in using it throughout the text.
**As mentioned above we consider GIA to include GRD effects. Gravitational pull is already included at the end of the sentence. We will omit "GIA-induced" and replace "bed deformation" with "solid Earth deformation".**
**We would rephrase as follows: "When ice sheets decay and grounding lines retreat, the viscoelastic solid Earth deformation reduces the water depth and hence induces indirectly a stabilizing negative feedback, which is additionally supported by near-field sea level lowering in response to reduced gravitational attraction (Gomez et al., 2010, 2012, 2015)."**

L51: "indirectly"
Why is this an indirect effect rather than direct?
**We will omit "indirectly".**

L53: "(Gomez et al. 2010, 2012, 2015)" There are studies that have specifically looked at the effect of gravitational attraction e.g., Coulon et al. 2021; Han et al. 2021 "
**We cited those three studies with regard to the sea-level feedback, of which the gravitational attraction is a relevant part. We will add the two more citations, as suggested by the reviewer.**

L56: "(lapse effect) "
lapse-rate effect
**Added.**

L61: "(Peltier, 2005)" Is this a right reference for LGM timing? I think it's okay to not have one here unless there's more generic one.
**Right, Clark et al. 2009 identified the LGM interval between 26.5 and 19 kyr BP. We will change this in the manuscript.**

L63: "Such inter-hemispheric "
The inter-hemispheric effect should be first clearly explained before saying "such" effect; the concept is not clear from the sentence before. You can make it clear that the "rapid sea level rise" since the LGM comes from the collapse of the ice sheet from the Northern Hemisphere.
**We agree and add: "A global sea-level low-stand was reached during the Last Glacial Maximum (LGM) between 26,500 and 19,000 years ago (26.5–19 kyr BP, Clark et al., 2009). The rapid sea level rise since**

**LGM due to the collapse of the northern hemisphere ice sheets likely contributed to the destabilization of the Antarctic Ice Sheet, as iceberg-rafted debris records in deep-sea sediments suggest (Weber et al., 2014; Jones et al., 2022), which accordingly is reflected in an inter-hemispheric synchronicity of the large ice sheets' changes (Gomez et al., 2020)."**

L69: "(Coulson et al., 2021) "
This reference focuses on crustal deformation in response to prescribed ice sheets in the Northern and Southern Hemisphere, so it is doubtful to be an appropriate one to cite in talking about the state of Antarctic Ice Sheet.
**Right, will be omitted.**
**"The present-day state of the AIS, including its contemporary rates of change, characterizes its stability and the potential for sea level rise in an increasingly warming climate (Joughin and Alley, 2011)."**

L71-83: "Ice sheet models have been coupled to GIA models with different levels of complexity …"
This paragraph should provide more literature review on the coupled ice sheet - GIA models with the highest complexity in the field. For now, it mainly focuses on the models with the lowest level of complexity with ELRA and LC, which do not incorporate gravitational and rotational effects. It would be only fair to add more review on those that already incorporate both gravitational and rotational effects as well as the 3D earth structure.

**In L84-91 we list more complex GIA models (where we added Han et al., 2021) capturing GRD effects, and 3D Earth structure in L94-97. We will also refer to the overview Table 1 from Swierczek-Jereczek et al., [in review].**

L71: "(de Boer et al., 2017; Whitehouse, 2018). "
I would recommend citing the studies that have done the actual work of coupling an ice sheet model and a GIA model rather than citing review papers that describe the state of the field. Most of the references that can be cited here are already mentioned elsewhere (i.e. Gomez et al. 2012;2015, deBoer et al. 2017; Konrad et al. 2015, etc.)
**For this general introductory statement, we believe, a review paper can provide a good overview for further reading. We added Swierczek-Jereczek et al., [in review] to this list. Further down in the paragraph we mention individual studies, who have done the actual work, as suggested. This avoids double citing.**

L82: "(Goelzer et al., 2020) "
This ref introduces a method on how to correct for sea-level external forcing in calculating icesheet contribution to sea level, which happens at the post-processing part, rather than using GMSL as external forcing to drive ISMs?

**Goelzer et al. 2020 provides a method for calculating global mean sea level contribution from regional ice sheet models, that usually are forced with external global mean sea level time series (not coupled), and this is what this sentence says.**

L83: "with polar motion "
Rotational effects need to be introduced earlier in the introduction section. **We will place a sentence following L39: "The redistribution of land ice, ocean water and mantle material induces changes in the Earth's gravity field (with the 'geoid' as equipotential surface), and therefore alters the sea level globally, which follows the geoid (Farrell and Clark, 1976). As a further feedback mechanisms, polar motion due to the surface mass redistribution changes the geoid and deforms the solid Earth (Mitrovica et al., 2005). This gravitational-rotational-deformational (GRD) correction is usually considered as being part of a consistent description of GIA."**

L91: "presumably "  doesn't seem necessary.
**Agree.**

L94: "Recently, a new generation of GIA models accounts for the 3D Earth structure (A et al., 2012; van der Wal et al., 2015; Nield et al., 2018; Powell et al., 2021; Blank et al., 2021) "
The sentence seems incomplete. Also, Latychev et al. (2005) developed a 3D GIA model, and Powell et al. 2021 used it. And how about the original references for VILMA?
**We would rephrase as follows: "Since the 2000s, 3D viscoelastic Earth models were developed (e.g. Martinec, 2000;  Latychev et al., 2005, Wu et al., 2005, Zhong et al. 2022, Huang et al., 2023) and considered in a new generation of GIA models (e.g., Klemann et al., 2008, A et al., 2012; van der Wal et al., 2015; Nield et al., 2018; Bagge et al., 2021; Powell et al., 2021; Blank et al., 2021; Yousefi et al., 2022)."**

L100-102: "Here, we present a set of new simulations of Antarctic Ice Sheet evolution over the last 246,000 years (i.e. two full glacial cycles) with the Parallel Ice Sheet Model (PISM) coupled to the VIscoelastic Lithosphere and MAntle model (VILMA) solving for GIA. "
It would be helpful to describe here in more detail how this study fits into the context of the work introduced above and in the field. For example, you mention above the interactive coupling scheme for 1D and 3D ISM-GIA models in the field, what do you similarly or differently? Is your goal to have a coupled model functionality that can iteratively converge towards an initial bed topography? Or come up with a coupling timestep that is the most appropriate to use over the glacial timescale/for a specific load response? What's the reason for doing your experiments for two glacial cycles?
**As the title of our studies suggest, we want to improve our understanding of `feedback mechanisms controlling Antarctic glacial cycle dynamics', which is for marine ice sheets particularly associated with grounding line dynamics. We show that we have the adequate tools to do so (description of the coupled model, 3D Earth structure, coupling interval, resolution, computational cost), with a degree of complexity at the upper end of the listed models. We use sensitivity tests (e.g. variation in Earth**

**structure and involved time scales) to disentangle feedback strengths and interactions. As we want to investigate in follow-up studies sea-level projections and tipping thresholds in a warming future climate (with coupling intervals of 1yr), this study describes the necessary framework and the model spin-up. We will provide a better motivation early on in the revised manuscript.**

L115: "interpolated accordingly "
According to what? Does this mean that basal melt is not calculated but rather taken from somehwere else predefined? Or do you mean the interpolation of basal melt at the ice draft?
**Basal friction and basal melt are interpolated according to the sub-grid grounding line interpolation scheme by Gladstone et al., 2010 mentioned in the previous sentence. As indicated in the cited literature (e.g. Seroussi & Morlighem, 2018), this provides a sub-grid scale `representation of basal melting at the grounding line in [coarse] ice flow models', and hence influences the melt sensitivity of the ice sheet model.**
**We rephrase as follows: "The grounding line location simply results from flotation condition, without additional flux conditions imposed (Reese et al., 2018c). Basal friction and basal melt are interpolated according to a linear sub-grid interpolation scheme (Gladstone et al., 2010; Seroussi and Morlighem, 2018; Hewitt and Bradley, 2023)"**

L134: "sea level "
 "sea-surface height" or "sea surface geoid" would be more precise term to use here.
**Changed to: "The LC model is computationally efficient, the default coupling time step is 10 yr, but it does not account for regional variations of the sea-surface geoid."**

L149: "rotational deformations "
Do you mean the effect of rotational feedbacks on viscoelastic deformation.
**Changed to: "Furthermore the rotational feedback is accounted for (Martinec and Hagedoorn, 2014, following the discussion in Mitrovica et al. 2005)." As we discussed this above, this sentence should be sufficient**

L150: "gravitational "  gravitationally?
**Of course.**

L153: "Coast lines can freely migrate according to the local RSL, also accounting for floating ice shelves "
What does it mean that coastlines migrate accounting for floating ice?
**Changed to: "Coast lines can freely migrate according to the local RSL. The loading effect of floating versus grounded ice is accounted for. The grounding line location is determined by the flotation conditions and hence by the densities of ice and ocean water (consistent with PISM)."**

L155: "redistribution of water mass" "exchange of mass"?
**We prefer "redististribution".**

L160:"condition, "
remove since core itself is a boundary rather than a condition?
**Changed to: "The interaction with the fluid core is considered as a boundary condition, …"**

L160: "we " is?
**OK**

L161: "n128"
describe what 'n' is.
**This is a technical specification and describes the number of co-latitutinal Gauss-Legendre points by two.**

L161: "256×512 "
If you are setting the spherical harmonics degree and order to be 170, shouldn't the grid resolution be 160X340?
**This is based on an alias-free condition, which defines the numbers of latitudial Gauss-Legendre points to be larger than 3/2 of the maximum degree considered for the solid earth (e.g., Martinec, 1989).**

L164: "gravity-consistent "
Have a consistent expression throughout the text - in Line 150, 'gravitational consistent' is used.
**Agree to change to "gravitationally consistent".**

L165: "mass redistribution of water "
Remove "of water".
**OK**

L168: "…for flexible restarts, the original Fortran77 code was modernized to Fortran90…"
What do you mean by "flexible" restart? Did (or not) VILMA have a restart functionally before but it was restricted with some technical issue? And how was the code modernized? Please be more descriptive.
**The original code had no restart capabilities, as it was not necessary before. We will rephrase: "In order to allow for restarts, the original Fortran77 code was modularised and in main parts extended using Fortran90 commands during the PalMod project (see acknowledgements) and OpenMP parallelization was implemented explicitly."**

L169: "calculation of the left-hand side elastic problem and the right-hand side viscous solution "
Don't see any left or right-hand side. Provide an equation or a reference to it?
**We have shortened it, see above.**

L169: "For this, openMP4 functionality "
Isn't OPEN MP for parallelization? If so, how does this relate to the restart functionality?
**See above.**

L175:"PISM and VILMA are offline-coupled using a coupling time step that can be rather short. This is an advantage of the explicit time stepping and the weak formulation (in time-domain) of the solid-Earth dynamics (Martinec, 2000) in contrast to the normal-mode approach (in Laplace domain) used by most of the 1D Earth models". The "coupling time step" in this sentence is the time interval after which PISM and VILMA call the other, and the explicit time stepping referred in this sentence is specifically for VILMA's time stepping. It could be helpful for readers to make connection between the coupling timestep and VILMA's timestep by explicitly mentioning that the coupling interval is constrained by the VILMA time stepping. In addition, the 1D normal mode approach also allows a short time stepping down to annual scale (e.g. Han et al. 2022). So the comparison made in this sentence might not be correct.

**We agree that this can be formulated more clearly and decided to skip the discussion regarding the limitations of the normal mode approach, as this is not relevant for the discussion here.**

**"PISM and VILMA are offline-coupled using a coupling time step that can be rather short and is only constrained by the explicit time steps chosen in VILMA and PISM for integration of the respective field equations. As default, we apply a coupling time step of $\triangle$t=100yr, at which PISM and VILMA exchange data.** ~~This is an advantage of the explicit time stepping and the weak formulation (in time-domain) of the solid-Earth dynamics (Martinec, 2000) in contrast to the normal-mode approach (in Laplace domain) used by most of the 1D Earth models (e.g. Peltier, 1974). The normal mode theory generally hampers a short time scale coupling, due to the fact that for each additional time step in principle the convolution over the previous loading history has to be repeated.~~"

Figure 2: "coupling time step "
Include the 'delta t' symbol in the figure.
**Good idea.**

L179: "…VILMA integrates "  "runs"?
**When VILMA runs, it integrates the field equations, so we prefer to use this phrase.**

L181: "VILMA interpolates "
What does this mean? Why does ice history need to be interpolated temporally if it's provided by PISM at every coupling interval?
**We decided to exchange the load distribution only at the coupling intervals, here synonymous to "snapshots". Between these two states, hence within the coupling time step, the load change is interpolated linearly. Of course, for an internally smaller time step in PISM output, we could also exchange the ice history between $t_{i-1}$ and $t_i$ at smaller time steps. The same is valid for a more-frequent VILMA output to PISM. If the coupled dynamics require a finer temporal resolution, we can easily reduce the coupling time step.**

L188: "and water load "
Remove or rephrase unless PISM actually knows about the global water loading distribution.
**Agree, this was an error in the description.**

L190: "the PISM response always lacks behind the VILMA step by one coupling step "
Explain what this means? PISM runs between t0 and t1, and only after then VILMA will run from t0 and t1, and so on; I take you are trying to say that the RSL change from t0 and t1 calculated in VILMA is only reflected in ice dynamics in PISM for t1 and t2, not from t0 and t1.
**Exactly, we rephrased: "In our coupling scheme, the PISM response always lags behind the VILMA step by one coupling step, i.e. RSL change from $t_{i-1}$ to $t_i$ calculated in VILMA is only reflected in ice dynamics in PISM from $t_i$ to $t_{i+1}$."**

L213: "(long) memory effects of the AIS to the climate history "
This is not the best reason for running two glacial cycles. Is there a better reason?
**For the GIA part 40 kyr spin-up may be sufficient to cover the memory for the load history. Ice sheets can have a much longer memory, particularly in their thermal state. For PISM simulations with LC and similar settings as in this study, we have shown this sensitivity of the present ice sheet state (and generally during deglaciation and interglacials) to the thermal spin-up several glacial cycles back (Albrecht et al., 2020a, Sect. 1.3), but same initial geometries. In our study, the pen-ultimate glacial cycle can be considered as "warm-up round" for the last glacial cycle, so we do not discuss these model results in more detail.  Furthermore, the preset glacial cycle accommodates the non-isostatic state of the Earth's mantle during the last interglacial and the following glacial inception.**

Figure 3 caption: "c) Ice volumes above flotation have been calculated considering cell-area weighting and density correction (Goelzer et al. 2020)… "
So you are correcting sea-level change with bedrock change correction term and density correction term from Goelzer et al. 2020? Any external sea-level forcing correction term? Also, include descriptions for the legend in the figure caption.
**Goelzer et al., 2020 describes a density correction (Sect. 5) and a bedrock change correction (Sect. 4) to the standard "volume above flotation" approach (Corrigendum), used for the post-processing analysis of regional ice sheet model changes, to derive global mean sea level contributions. In the coupled PISM-VILMA simulations, bed elevation changes are fully described by RSL changes from VILMA relative to the geoid (RSL=z0-bn, z0=0), so no external sea-level forcing is used. We do not apply the bed elevation change correction (water expulsion effect), as RSL from VILMA does not distinguish between contribution from viscoelastic bed elevation changes or from regional sea surface height patterns, such that the far-field RSL change would counteract the near-field RSL changes (on a global grid the sum would vanish).**
**In the revised manuscript, we will state explicitly, what conversion we have used, and which corrections we have not used and why. The figure caption will be updated referring to the legend.**

Figure 3c:

Any idea what's the reason for a big difference in 3D min and max in panel c?

**We discuss this difference in convergence rates in L427. This effect of alternating sign in mean RSL anomaly, particularly in the Siple Coast region in the Ross Sea sector, is shown in Fig. 4 for 3Dref, which features a similar convergence rate as 3Dmin. The same effect has previously been discussed in van Calcar et al., 2023, where an average of the last iterations steps is used for the final iteration. This may help for shorter internal iterations, but it did not help for longer external iterations.**

Figure 4.:

Is this taking what is labeled as "3D ref" in Fig 3? State it in the caption. Also, please describe the black and grey contour lines.

**Sure.**

L235: "previous PISM simulations"

Previous PISM studies?

**OK**

L247: "The '3D min' and the '3D max'… "

This introduction needs to come when Fig. 3 is first introduced. Also, by looking at the profile in the supplementary figure, these two profiles are laterally homogeneous, and more like 1D Earth profile? If so, would be helpful to state that.

**We only added the 3Dmin and 3Dmax results in this Fig. 3 to avoid a further figure. We will refer to Sect. 2.4. in the caption of Fig. 3.**

**The structures are 1D Earth profiles only below Antarctica (most of the Antarctic region covered in the figure), while the rest of the globe remains 3D. We state this in L249.**

L238 "…viscosity of 5×1020 Pa s). However, this approach is limited to a certain region (e.g. Antarctica) and is unable to solve self- consistently for sea level changes while considering a globally conserved water budget."

This information might be better combined with the part in Introduction where it talks about the limitations of solid earth models with simple structure and no gravitational and rotational effects. It sounds little redundant and out of place here.

**We will omit L235-239 , values for mantle viscosity and elastic lithosphere are visible in Fig. 5a, L240 is then redundant, so we start following paragraph with "Between [...] "**

L261: "sea level equivalent ice volume (in units of mSLE) "

Please describe or provide an equation how this calculation is made. Is this based on VAF without correcting for bedrock elevation change and/or density difference between fresh water and ocean?

**As mentioned above, in our analysis on the PISM grid, we use the VAF approach for better comparison to previous model results, with density correction but no bedrock elevation correction, while the ocean area considered in the conversion is constant.**

L265: "(blue) " blue lines in Fig. 6

**Agree**

L266: "… 1000 yr delay. "
I might be confused - Why is this considered a 'delay'? Between t1 and t2, PISM is accounting for RSL changes between t0 and t1 calculated by VILMA, so at t1 PISM is not seeing any "delay" in RSL change that happened between t0 and t1?
**This is true, but the previous changes in bed elevation (RSL from VILMA) remain fixed over the full coupling time step in PISM, which also is considered in the VAF to estimate the inferred sea-level change. We chose the 1000-yr time step here only to make this log-in effect better visible.**

L266-267:"This is mostly a consequence of the diagnostic calculation of the sea level equivalent ice volume (above flotation) that "
Why would the diagnostic calculation of the SLE VAF influence on the discontinuity, especially just for the 1000yr coupling interval? Even if so, would that be the entire reason? The timing of the appearance of the most pronounced discontinuity seems to coincide with the when Melt Water Pulse 1A event during which so much of the ice sheet melted from the Northern Hemisphere. I wonder if the AIS from PISM experienced a large far-field sea-level feedback from the Northern Hemisphere over each coupling interval, which might have influenced the ice sheet to be more unstable and thus affect the shape of the curve? I'm thinking of the mechanism introduced in Gomez et al. 2020. To check if this really an artifact of the SLE calculation, it would be helpful to show the plot just on ice volume or mass that is a raw output of PISM with no post-processing.
**The calculation based on VAF yields a signal in sea-level contribution also in cases where no ice thickness changes but the marine bed elevation (or sea level) does. This was the motivation for the corrections in Goelzer et al., 2020 and Adhikari et al., 2020. Hence, the inferred time series based on VAF in Fig. 6a (blue solid line) would be much smoother when considering their corrections, or when considering total ice mass or total ice volume (*see Fig. AC8*), instead.**

[Figure]

***Fig. AC8: AIS total ice mass change over last 20 kyr from coupled simulations with 3Dref Earth structure and different coupling time steps.***

**The MWP1a is a major driver of Antarctic deglaciation in our simulations (not necessarily a feedback here, but in line with Gomez et al., 2020), also visible from Fig. S4. Of course, the wiggles are most pronounced during largest RSL changes (about 20 m over 350 years in case of MWP1a), which aggregate to larger amounts for longer coupling time intervals. Antarctic deglaciation is highly sensitive to boundary conditions (Albrecht et al., 2020a). Of course the staggered forcing may influence the pathway of deglaciation, even though the blue solid curve in Fig. 6a varies only after about 11 kyr BP, while the dashed blue line with linear extrapolation diverges earlier (almost without wiggles). For our study, it is important that differences remain small when using 10-yr instead of 100-yr coupling time steps.**

L267: "… also accounts for changes in bed elevation. "
I see. And density correction? Also, provide reference - Goelzer et al. 2020?
**See comments above.**

L269: "(blue dashed) "
blue-dashed line in Fig. 6
**Agreed**

L269-270 & 278: "Comparing the ice sheet response for 100-yr (default) and 10-yr coupling time steps, we find only little differences, which we interpret as 'convergence' "
I think it is important to put the convergence your results within the context of spatial resolution of both PISM and VILMA, as it must depend on both.
**We agree with the referee that spatial and temporal resolution of grounding line dynamics are linked (similar to the CFL criterion in numerics). As ratio of a typical (elastic) response distance and a given typical grounding line migration rate, we may infer a measure for a maximum coupling interval.**
**In the manuscript we will provide more detailed information on the spatial resolution: "...for the given spatial resolution of n512/n128 in VILMA and 16km in PISM…"**

L276: "accurate "
precise, rather than accurate.
**Accuracy is a term used in numerics for convergence tests. We think this term fits ok here.**

Figure 6a:
What is 'dbdt' in blue dashed line in subplot a)?
**Right, "dbdt" is the rate of bedrock elevation change used for the extrapolation. The caption says "Lower temporal resolution with a 1000 yr is shown in blue, and with extrapolation of RSL change rate in blue dashed." We will change the legend to "expol".**

L285: "blue and black dashed lines"

blue solid and black dashed lines
**Agree**

L284-286: "A coarser spatial resolution in the sea level equation, however, delays deglaciation during the Holocene (blue and black dashed lines for n64 and n128, respectively), with up to 2m SLE difference to the reference resolution. "
Any idea why would it be this way?
**We would guess that the stabilizing sea-level feedback near the GL is overestimated for such coarse resolution of 0.7° or 1.4°, which amounts to 78 x 25 km or 156 x 50 km resolution at 71°S (*see Fig. AC9*), respectively, much coarser than the ice model grid of 16 x 16 km.**

[Figure]

*Fig. AC9: Difference in RSL at 10 kyr BP in Western Antarctica for different spatial resolutions of the VILMA sea level model, from left to right n64-n512, n128-n512, n256-n512 and n1024-n512, respectively. Largest differences of GL position and RSL anomaly are found in the Ross Sea region, explaining up to 1 m SLE (cf. Fig. 6b).*

Figure 6:
The black solid line in panel b) and a) should be equivalent based on the description (3D n512_n128 & dt 100 yr), but they don't seem to be. Also the black solid line in panel b) shows some weird curvature around 11kyr BP, jumping away from convergence. Is this why is this happening? As it is no clear pattern of convergence towards the black solid line in Fig. 6b, I wonder what's the rational for the authors decision to take this run (3D n512 n128) to expand onto doing sensitivity test to coupling interval. I suggest authors to re-check their plots (including the consistency between the black solid lines in Fig. 6a and Fig.6b) and re-do the experiment if necessary.
**The black solid lines in Fig. 6a and b do show results with the same resolution, but different initial conditions. As the figure captions says, in Fig. 6a " all simulations were restarted from the default state at 20 kyr BP (6th iteration) with default spatial resolution", consistent with all other figures, while in Fig. 6b  "All simulations have been initiated from the default state at 246 kyrBP (4th iteration)", for historical reasons.  We agree that it would be better to have a shared reference in both plots, but we cannot restart from 20 kyr BP VILMA restart state (6th iteration) with different spatial resolutions. Hence we would need rerunning the 10yr coupling interval over two glacial cycles, which would mean**

quite some additional computational costs and time. As the results within each panel are consistent, we believe a relative comparison is valid.

From Fig. 3b it can be inferred, that the fourth iteration with 3Dref reveals a speed up in deglaciation around 11 kyr BP, as the reviewer points out. We want to highlight that we have a highly non-linear coupled system with a high sensitivity to initial conditions. Convergence should be considered with respect to a certain range of internal variability (also discussed in Albrecht et al., 2020a). And this is also something we can learn from these results.

L290: "ice-dynamical effects for variation of the underlying Earth structure, "
You mean to say "effects of lateral variations in the underlying Earth structure on ice dynamics"?
**Agree**

L295: "mio. "
Describe what this is.
**This was supposed to be an abbreviation for million or $10^6$. Has been changed, see reviewer comment RC1.**

L313: "e.g. by viscoelastic bedrock uplift"
Include the other effects too since sea level change happens combination of changes in the sea surface height and bedrock height.
**Changed to: "... and partly compensates for near-field sea level drop in Antarctica, e.g. by viscoelastic bedrock uplift or gravitational pull..."**

L319: "light orange shading, "
the orange shading is not just far-field effects but combination with near-field effects, but it sounds like the light orange shading only represents far-field effects.
**In the far field, with distance to the AIS, i.e. in the corners of the map view, the near-field RSL effects are small compared to the far-field RSL effects, but there is no clear separation line, of course. We changed to: "...indicated by the light-orange shading towards the corners of the maps in Figs 8g, i and Figs. S 2c, d"**

L320: "cf. Wan et al., 2022 "
Wan et al. explores only the GIA model, not coupled ISM-GIA model. The original coupled modeling studies that saw this effect would be the ones to cite here. Also, make the use of 'cf' consistent throughout the text.
**This citation was referring to the start of the sentence "This quick and rather localized response", where we shifted the citation: "This quick and rather localized response (e.g., Wan et al., 2022) defines a negative and hence stabilizing feedback on grounding line retreat (Gomez et al., 2010, 2012)."**
**cf. means "compare to", we will check for consistency?**

L321:"rates of ice volume change and grounding line retreat remain comparably high until present "
But it's mentioned earlier the ice volume (SLE) has slower decline during the deglaciation phase.
**Right, this is no contradiction, as for a certain amount of ice loss, a slower decline means that, until present, stronger ice mass changes are possible. If it declines faster, especially during onset of deglaciation, there is less mass loss left at present. We would rephrase as follows: "For '3D min', rates of ice volume change and grounding line retreat remain almost constant at this comparably low level until present ."**

L341:"'3D ref' shows a trajectory in between those two end members "
This does not seem to be true for the post-LGM deglaciation phase, and the reason need to be discussed clearly in the text.
**This would be the hypothesis, what one would expect from the experiment. We rephrase: "Counterintuitively, the ice sheet response for '3D ref' does not show a trajectory completely in between those two end members."**

L343:"characteristics due to the weak Earth structure ('3D min') seem to dominate the overall ice sheet response, … "
Describe what this means.
**This was a rather qualitative assessment. The reasoning with regard to involved feedbacks is discussed later on. We would rewrite this sentence as:**
**"Yet, the coupled simulations reveal that in glacial periods of grounding line advance the ice volume trajectory evolves in a similar way as the trajectory for the weak Earth structure ('3D min'), while in deglacial periods of grounding line retreat the AIS volume responds in a similar way as the trajectory for the stiffer Earth structure ('3D max')."**

L366: "Antarctica"
Represented in the simpler Earth model.
**We add it.**

L366: "which also alters the global mean sea level "
Little confusing what "which" refers to and what this sentence means as a whole, regardless.
**This was a reference to Fig. 9a. We omit this part of the sentence and refer to the figure in the previous sentence.**

L365: "7mSLE"
This seems rather a huge difference (around 10% of the total LGM SLE). The difference might be reduced if the 1D model incorporated a lower upper mantle viscosity (($10^{18}$ or $10^{19}$ PaS), which is a better approximation of the viscosity in the West Antarctic region.
**Right, it is likely that the difference is mainly due to the stiffer mantle viscosity, as the coupled simulations with 3Dmin vs. 3Dmax suggest.**

L372: "In our coupled simulations we focus on the dynamics of the AIS in response to the northern hemisphere ice sheet's decay.
Add "For this section,".
**Agree**

L376"weak structure "
weak Earth structure
**Agree**

L381: "a synchronous coupling. "
If you mean "synchronous" as the ISM and GIA model communicate every coupling time step, Gomez et al. also uses synchronous coupling. Also, the coupling scheme presented in this work doesn't seem to be different in Gomez et al.'s.
**This statement refers to the described new coupling method in Gomez et al, 2018, where 3D GIA model and ice sheet model are run subsequently for the full 40 kyr, and iteratively converge to a coupled solution. Technically speaking, the coupling interval there would be 40 kyr, even though the temporal resolution of the exchanged data is 200 years.**
**The reviewer is right, that Gomez et al., 2020 uses the older coupling scheme from Gomez et al., 2013, where data between 1D GIA and ISM are exchanged synchronously every 200 years, as the method section indicates. We will omit "synchronous coupling" in the manuscript.**

L381: "we run iterations over the last two glacial cycles (246 kyr instead of 40 kyr). "
Given that the LGM deglaciation signal would be so dominant for the deglacial period (postLGM), how might running over two glacial cycles improve your results compared to running one glacial cycle from the LGM peak?
**As mentioned above, we find a much longer memory in the ice sheet's thermal state than in the Earth response. Furthermore, we show in our simulations that different feedbacks on GL migration dominate during glacial build-up and deglaciation depending on the Earth structure. Hence, LGM ice volume depends on that interplay and deglaciation depends on the LGM ice volume. Running only from the LGM peak would completely change the result, or it would only provide a limited focus on feedbacks during deglaciation. We will add some more discussion on this path dependency in the revised manuscript.**

L389: "As barystatic mean sea level change since LGM we find 120m (about 107m from the northern hemisphere in ICE-6G_C) "
It's already mentioned in Line 383 that 13m SLE is seen from your simulation.
**Right, we will omit the 107 m, even though this is a good consistency check, as the 13 m were calculated on the PISM grid based on the AIS changes, while the 107 and 120 m were averaged over the globale RSL changes, with constant AIS and with evolving AIS, respectively**

L390: "we prescribe the present-day AIS configuration, "

do you mean when you "fix" the AIS through the GIA simulation and just see the fingerprint of the NH ICE6GC on Antarctica?

**Yes**

L378: "ICE-5G, "

Gomez et al. 2020 also did sensitivity test using different NH ice histories using ICE6GC.

**Right, but the plots (Fig.1 and 3), we compare to, were based on ICE-5G.**

L393: "viscoelastic "

viscoelastic deformational

**Changed to: 'deformational'.**

Figure 10:

The figure captions are almost identical to the one from Gomez et al. I don't know the rule for when you are reproducing plots from other papers, but you should make sure you are not plagiarizing by accident.

**We state in the caption title 'analogous to Gomez (2020), Fig. 1'. For comparison it makes sense to be as close as possible to the original Fig. 1, This is also valid for the figure caption, which helps to identify differences. We will reformulate the caption and discuss differences in the main text, and we will make clear at the caption end that we used here a similar formulation for better comparison: "Figure design and caption has been chosen similar the original for better comparison."**

Figure10:"analogous to Gomez et al. (2020) "

Include the figure number from Gomez et al. as well?

**Sure.**

L395: "forebulge effect "

The last time this term appears is in the abstract, and it would be helpful to have a clear description on what this is before using this term.

**L330-333 introduces the forebulge as "...while in the adjacent coastal region the sea floor becomes about 50 m shallower than for '3D max' (Fig. 8a, light blue shading). This feature is called 'forebulge' and results from lateral transport of displaced mantle material and the flexure of the lithosphere." We will rather mention it as a feature rather than the "forebulge effect" here (see comment below), as the actual forebulge feedback on GL advance will be introduced later on. We will try to guide the reader more clearly to this effect through the revised manuscript.**

L396: "This effect is presumably smaller "

Possibly because the ice mass loss in Gomez et al. is much smaller than the simulation shown in this work. (referring to the comparison made in L383).

**Agree, we will state that: "At the edge of the continental shelf, a peripheral forebulge developed of opposite sign with up to 85 m vertical bedrock displacement since LGM. This feature is presumably smaller and hence not visible in the corresponding plots by Gomez et al. (2020), likely as a result of the much smaller AIS losses"**

Figure 8:

Describe the contour lines

**They are described in the caption, we added: "The modeled grounding line position due to the different Earth structures are shown as contour lines for '3D min' (orange), '3D max' (blue), '3D ref' (black) and as observed today (grey)."**

L428: "present-day"

Remove "-"

**Agree**

L428: "lowest convergence rates "

you mean the "lowest number of iteration" required to converge modeled present-day topography to observation?

**No, that would be the largest number of iterations required, when using this interpretation of that term. But in the first iterations one could also calculate a convergence rate from the ratio of RMSE of subsequent iterations, i.e. the relative reduction in RMSE by an additional iteration. This is usually done in numerical convergence tests. This approach fails, however, in case of alternating convergence (e.g. 3D min or 3D ref). We will add an explanatory sentence.**

L430: "The maximum LGM extent is very similar to the ... "

Inferred from what?

**Inferred from the grounded ice sheet area.**

L432: "then "

Than

**Agree**

L433:"GIA feecback"

sea-level feedback seems to be a more precise word to use because the definition of GIA in this work seems to only represent the deformational part.

**We agree, that sea-level feedback is more often used in literature. The term "GIA" in this study, however, also comprises GRD effects, as mentioned above.**

L450:"…as sufficient for capturing the relevant dynamics and feedbacks in glacial cycle simulations." Sufficient seems to be quite a vague word and maybe misleading. While it's true you are able to see differences in coupling vs. no-coupling with the coarse resolutions, it would be important to acknowledge the importance of high-resolution for capturing accurate ice dynamics at the grounding lines shown in many modeling studies.

**Changed to: "For coupling time steps of 100 yr between PISM and VILMA, grid resolutions of 16 km for PISM in Antarctica, 0.2° (≤20 km) for solving the sea level equation and 0.7° (≤78 km) for solving the viscoelastic deformations in VILMA, we are able to capture relevant dynamics and feedbacks in glacial**

**cycle simulations, acknowledging that the accurate representation of the complex grounding line dynamics would require spatial resolution of the order of 1km."**

L457: "higher resolutions are possible "
Is this true for the ice model as well?
**Of course, but this comes with about 10 times higher costs for resolution doubling. And, increasing the spatial resolution requires a re-tuning of other "resolution-dependent" model parameters, which is usually done with an ensemble approach.**

L464: "slow response "
More appropriate expression would be something like "slows the retreat rate of the ice sheet."
**We had "slow response of the retreating ice sheet", but we will change it accordingly.**

L467: "which can even cause grounding line re-advance. " But shouldn't this be the other way around because on slow uplift on retrograde-sloped bed would leave the MISI to keep acting.
**Yes, this is correct as long as the topography is predominantly retrograde, but the stabilizing effect on the GL comes with quite some delay, eventually leading to a grounding line re-advance. We added ", after the grounding line has stabilized at topographic features.".**

L451:"outer iteration "
This word has not been used in the text. I would suggest writing "topographic correction" or "iterative correction for initial topography" or some sort that is much more clear.
**We will use the latter suggestion.**

L477: "This provides confidence in the application of the PISM-VILMA coupling framework to future ice sheet and sea level projections and the investigation of tipping point characteristics "
This is a rather strong statement to make. Ice-sheet models that focus on predicting future sea level emphasize the importance of high-resolution and higher-order velocity solvers for capturing rapid evolution at grounding lines as well as small-scale peripheral glaciers. Plus, the work by Albrecht et al. 2020b. uses the same ice-sheet model, PISM, so I'm not sold by this argument. As mentioned in a comment above, the importance of high-resolution ice sheet modeling for grounding line dynamics and the short-coming of the model used here need to be acknowledged.
**PISM has been used in the ISMIP6 and ISMIP6 extensions both for Greenland and Antarctica (Seroussi et al., 2020) as well as in high resolution (up to 0.45 km) sea level projections for Greenland (Aschwanden et al., 2018), 1 km for Antarctica is technically feasible. PISM can account for higher-order stress balance (Blatter-Pattyn), the resolution aspect for adequately capturing grounding line dynamics is now acknowledged in the text above. PISM-LC has been already used in long-term sea-level commitment and hysteresis analyses (Garbe et al., 2019, Klose et al. 2023). The choice of PISM settings and resolution in this study according to Albrecht et al., 2020a,b serves a better comparison of involved feedbacks and benefits from the previous work (constraining parameters). In light of the gained experience with the more complete GIA model VILMA applied to Antarctic glacial**

cycles, we find simulations for future scenarios as the logical consequence. Feedbacks and tipping point characteristics require millennial-scale simulations and could directly profit from the spin-up done in the study, as long as the system understanding rather than exact numbers would be the focus. In case of centennial scale projections, of course we will use higher spatial resolution (4 or 2 km) and higher coupling frequencies of 1 yr, capturing grounding line dynamics accurately. Yes we have that confidence and experience and we believe we can make this strong statement.
* * *
References:

Coulon et al 2021. Contrasting Response of West and East Antarctic Ice Sheets to Glacial Isostatic Adjustment

Latychev et al. 2005. Glacial isostatic adjustment on 3-D Earth models: a finite-volume formulation

Han et al. 2021. Modeling Northern Hemispheric Ice Sheet Dynamics, Sea Level Change, and Solid Earth Deformation Through the Last Glacial Cycle

**References, which were not already listed in the original manuscript:**

**Clark et al. 2009**          **https://doi.org/10.1126/science.1172873**

**Han et al., 2021**          **https://doi.org/10.1029/2020JF006040**

**Hewitt and Bradley, [preprint]**  **https://doi.org/10.21203/rs.3.rs-2924707/v1**

**Huang et al., 2023**          **https://doi.org/10.1093/gji/ggad354**

**Konrad et al., 2014**          **https://doi.org/10.1007/s10712-013-9257-8**

**Latychev et al., 2005**          **https://doi.org/10.1111/j.1365-246X.2005.02536.x**

**Lloyd et al., 2020**          **https://doi.org/10.1029/2019JB017823**

**Martinec, 1989**          **https://doi.org/10.1016/0010-4655(89)90043-x**

**Mitrovica and Peltier, 1991**      **https://doi.org/10.1029/91JB01284**

**Mitrovica and Milne, 2002**    https://doi.org/10.1016/S0277-3791(02)00080-X

**Mitrovica et al., 2005**    https://doi.org/10.1111/j.1365-246X.2005.02609.x

**Stocchi et al., 2013**    https://doi.org/10.1038/ngeo1783

**Swierczek-Jereczek et al. [preprint] https://doi.org/10.5194/egusphere-2023-2869**

**Wu 2004**    https://doi.org/10.1111/j.1365246X.2004.02338.x

**Wu et al., 2005**    https://doi.org/10.1016/j.jog.2004.08.006

**Yousefi et al., 2022**    https://doi.org/10.1029/2021GL097525

**Zhong et al. 2022**    https://doi.org/10.1029/2022GC010359

---

## Author Response (AR1)

**Author's Response**

A detailed point-to-point response to all reviewer's questions and comments have been uploaded as Author Comments

AC1: https://doi.org/10.5194/egusphere-2023-2990-AC1 and
AC2: https://doi.org/10.5194/egusphere-2023-2990-AC2.